



# Environmental and hydrologic controls on sediment and organic carbon export from a subalpine catchment: insights from a time-series

Melissa S. Schwab[1,2], Hannah Gies[1], Chantal V. Freymond[1,3], Maarten Lupker[1,4], Negar Haghipour[1,5], Timothy I. Eglinton[1]

[1]Department of Earth Sciences, ETH Zurich, Zurich, 8092, Switzerland
[2]Now at Jet Propulsion Laboratory, California Institute of Technology, Pasadena, 91109, USA
[3]Now at Gruner, Basel, 4020, Switzerland
[4]Now independent, Bern, 3014, Switzerland
[5]Laboratory of Ion Beam Physics, ETH Zurich, Zurich, 8093, Switzerland

*Correspondence to*: Melissa S. Schwab (melissa.s.schwab@jpl.nasa.gov)

**Abstract.** Studies engaging in tracking headwater carbon signatures downstream remain sparse, despite their importance for constraining transfer and transformation pathways of organic carbon (OC) and developing regional-scale perspectives on mechanisms influencing the balance between remineralization and carbon export. Based on a 40-month time series, we investigate the dependence of hydrology and seasonality on the discharge of sediment and OC in a small Swiss subalpine

watershed (Sihl River basin). We analyze concentrations and isotopic compositions ($\delta^{13}C$, $F^{14}C$) of particulate OC and use dual-isotope mixing and machine learning frameworks to characterize and estimate source contributions, transport pathways, and export fluxes. The majority of transferred OC is sourced from plant biomass and soil material. Relative proportions of soil-derived particulate OC peak during the summer months, coinciding with maximum soil erosion rates. Bedrock-derived (petrogenic) OC abundant in headwater streams progressively decreases downstream in response to a lack of source material

and efficient overprinting with biospheric organic matter, illustrating rapid OC transformation over short distances. Large variations in isotopic compositions observed during baseflow conditions converge and form a homogenous mixture enriched in OC and characterized by higher POC-$F^{14}C$ values following precipitation-driven events. We propose that storms facilitate surface runoff and shallow landsliding, resulting in the entrainment of fresh litter and surficial soil layers. Model results further indicate diverging mobilization pathways. Discharge and water stage describe the export of suspended sediment,

while the prediction of POC fluxes is mostly supported by water stage and 1-day antecedent precipitation. Although particle transport in the Sihl River basin is mainly driven by hydrology, subtle changes in bedrock erosivity, slope angle, and floodplain extent likely have profound effects on the POC composition, age, and export yields.

## 1 Introduction

River networks serve as a continuum ultimately connecting the terrestrial with the marine biosphere (Aufdenkampe et al.,

2011). Disproportional to their spatial extent, water bodies are active sites for transport, transformation, and storage of



significant portions of organic carbon (OC) mobilized from the terrestrial environment (Battin et al., 2009). Annually, between 1.90 and 2.95 PgC yr$^{-1}$ are entrained into inland waters (Cole et al., 2007; IPCC, 2021; Tranvik et al., 2009; Regnier et al., 2022). The majority of this carbon is lost during transfer due either to remineralization and outgassing or to burial in lakes and floodplains. Ultimately, only 0.80-0.95 PgC yr$^{-1}$ reach marine coastal regions (Tranvik et al., 2009; Battin et al.,

2009; Raymond et al., 2013; Lauerwald et al., 2015; IPCC, 2021; Regnier et al., 2022). However, anthropogenic and climate-driven changes markedly influence erosional processes and thus may perturb the translocation and sequestration of OC in freshwater systems. The human-induced lateral transfer of carbon adds ~0.60 PgC yr$^{-1}$ to inland waters which are largely respired and buried during fluvial transit (Regnier et al., 2022; Lauerwald et al., 2020; Li et al., 2019). The resulting OC flux to marginal shelves deviates by only 0.15 PgC yr$^{-1}$ from pre-industrial values (Regnier et al., 2022).

It is well established that small mountainous rivers deliver substantial quantities of sediment and particulate organic carbon (POC) to the oceans (Milliman and Syvitski, 1992; Lyons et al., 2002; Leithold et al., 2006; Hilton et al., 2012; Goñi et al., 2013). These systems are often characterized by a steep basin morphology, with little to no developed floodplains. The resulting basin storage capacity is insufficient to retain the large amounts of eroded sediments and soils exported from mountainous catchments (Wheatcroft et al., 2010; Milliman and Farnsworth, 2013). During storm-driven events,

mountainous rivers are strongly coupled to hillslope processes and often provide the main source of sediment to downstream channels (Milliman and Syvitski, 1992; Hilton et al., 2012).

A decrease in the OC content of the suspended load is commonly observed with increasing discharge in high sediment-yield small mountainous rivers (Masiello and Druffel, 2001; Coynel et al., 2005; Leithold et al., 2016). This decline corresponds to the dilution with carbon-poor bedrock material. Less well constrained is the character of the exported POC. The proportion of

fossil OC concomitantly increases with rising sediment yields in small river systems as they share a common source and transport pathways (Blair et al., 2003; Komada et al., 2004; Leithold et al., 2006; Hilton et al., 2011a). While deep-seated landslides and gully erosion mobilize predominantly bedrock-sourced (petrogenic) OC, surface runoff and shallow landslides preferentially remove fresh litter and organic-rich surface soils (Hovius et al., 2000; Hilton et al., 2008b; Hatten et al., 2012; Goñi et al., 2013).

In headwaters, significant portions of biospheric organic matter are exported in the form of coarse POC (>1 mm) encompassing leaves, needles, and wood fragments (Turowski et al., 2016; Rowland et al., 2017). In contrast to fine-grained OC, which can remain in suspension for prolonged periods, coarse particles are often deposited in headwater valley segments due to gravitational settling or retention in log jams (Wohl et al., 2012; Jochner et al., 2015). Interaction of woody debris and the gravel bedload might lead to grinding and size reduction, ultimately adding to the fine POC pool (Turowski et al., 2016).

Despite these mechanisms, restricting the transport of coarse POC, studies showed that event-driven floods can effectively recruit and transfer vascular plant debris as driftwood (West et al., 2011; Wohl and Ogden, 2013; Wohl, 2017; Ruiz-Villanueva et al., 2019) or as a component of the suspended load (Schwab et al., 2022) to continental margins.



Despite the growing recognition of the substantial and rapid downstream transfer of terrestrial OC from headwater streams (Leithold et al., 2016; Wheatcroft et al., 2010; Goñi et al., 2013) and of the processes controlling organic matter during its

transfer through lowlands and floodplains (e.g., Bouchez et al., 2010; Hemingway et al., 2017; Repasch et al., 2021), river segments connecting small, mountainous streams with lowland system remain poorly explored. A few studies address temporal dynamics of POC export in moderately steep river basins spanning timescales of individual storm events to intra- and inter-annual variability (Smith et al., 2013; Hatten et al., 2012). Even fewer studies examine the downstream evolution, composition, and molecular signature of OC in dynamic mountainous river systems (Goñi et al., 2014).

The focus of this study is to investigate in detail the response of sediment and bulk OC to variability in seasonality and discharge behavior in a moderately steep river basin bridging the gap between headwater streams and lowland rivers. The sub-alpine Sihl River links small mountainous headwater streams with the higher-order river Limmat, providing a crucial window on downstream transport and the evolution of OC along the riverine continuum. We obtained a high-resolution time series over 40 months focusing on the content, composition ($\delta^{13}$C, F$^{14}$C), and flux of sediment and POC. Export fluxes are

modeled using traditional and machine learning approaches, while a dual isotope model framework allows the estimation of potential organic matter source contributions. We discuss control mechanisms regulating organic matter mobilization and transport and examine our results in the context of previously published data on Sihl River headwater catchments in order to derive comparisons regarding the nature of exported POC (Smith et al., 2013; Turowski et al., 2016; Gies et al., 2022).

## 2 Methods

### 2.1 Characteristics of the Sihl River watershed

The Sihl River basin located in the Swiss Prealps is part of the Rhine River headwater system (4$^{th}$ order tributary; **Fig. 1**). Its watershed covers an area of 346.0 km$^2$ ranging from an elevation of 1872 m (Druesberg) at its headwaters to 402 m at the catchment mouth, with an average slope of 19.5º (**Fig. 1a-b, Table 1**). The Sihl River basin experiences a humid continental climate with a mean annual air temperature of ~9.5ºC and annual precipitation varying from 1450 to 1830 mm, with snowfall

generally occurring between November and April (MeteoSwiss, https://gate.meteoswiss.ch). The sub-alpine catchment is divided into an upper (145.7 km$^2$) and lower basin (190.1 km$^2$) near Einsiedeln by the reservoir Lake Sihl (10.2 km$^2$) constructed in 1937 (Addor et al., 2011). The damming of Lake Sihl results in the abrupt fragmentation of the flow path and the effective capture of sediment entrained from the upper watershed. Further regularization structures such as weirs and the partly channelization with concrete embankments strongly diminish fluvial connectivity supporting the retention and

trapping of 62-67% of sediment in the lower Sihl River basin (Grill et al., 2019). The Alp and Biber River tributaries are free-flowing rivers and major sources of water and sediment to the Sihl River. Similar to other small mountainous river systems (Wheatcroft et al., 2010), the steep morphology of the watershed and the absence of extensive floodplains limits the water storage capacity of the Sihl River. In response to severe storms, discharges can rise from an average of 6.8 m$^3$ s$^{-1}$ to



over 200 m$^3$ s$^{-1}$ and result in devastating flash floods (e.g., August 2005: 280 m$^3$ s$^{-1}$; Bezzola and Hegg, 2007; Jaun and
Ahrens, 2009).

The land cover of the Sihl River basin in 2017 consisted of 38.9% meadows and pastures, 9.0% urban settlements, 6.1%
unproductive areas, 1.3% water bodies, and 1.1% cropland (**Fig. 1c**; Federal Statistical Office, https://www.bfs.admin.ch),
with the majority of the watershed covered by forests (43.4%; Waser et al., 2017). The main tree species in the upper basin
are spruce (*Picea abies*) and fir (*Abies alba*) which are gradually replaced by deciduous trees such as beech (*Fagus*
*sylvatica*), maple (*Acer spp.*), ash (*Fraxinus spp.*), and oak (*Quercus petraea*) in the lower basin (Schleppi et al., 2006).
Forest soil carbon projections estimate stocks between 5 to 20 kgC m$^{-2}$ (Nussbaum et al., 2014; van der Voort et al., 2019).

The lithology of the lower Sihl River basin is composed of weakly consolidated, clastic sediments of the Molasse basin (**Fig.
1d**; Swisstopo; https://www.swisstopo.admin.ch). Limestones and other biogenic sedimentary rocks are common in the
allochthon nappes, Klippen, and Säntis zones. Campanian-Maastrichtian and Eocene flysch (Schlieren and Wägital flysch),
largely composed of mudstone and calcareous sandstone, predominate in the Alp and upper Sihl River basins (Winkler et al.,
1985). Dolomitic rocks of the northern limestone Alps and metamorphic rocks of the Arosa zone outcrop in the southern region
of the upper Sihl River basin. Stream valleys and lowlands are filled with unconsolidated rock material such as alluvions,
moraines, and gravel deposits that constitute important groundwater aquifers (Doppler et al., 2007).

The Erlenbach, Lümpenbach, and Vogelbach (**Fig. 1a**) are monitored as experimental catchments by the Swiss Federal
Institute for Forest, Snow, and Landscape (WSL), and are well studied in terms of terrestrial OC sources, mobilization, and
export of fine and coarse material (Schleppi et al., 1998; Hagedorn et al., 2001; Turowski et al., 2011, 2016; Rickenmann et
al., 2012; Smith et al., 2013; Gies et al., 2022; Hilton et al., 2021). These streams are second-order tributaries to the Sihl
River, with catchment sizes ranging from 0.7 to 1.6 km$^2$ and average discharges of 38 to 77 L s$^{-1}$ (Smith et al., 2013; Gies et
al., 2022). The Erlenbach basin is developed on an extensive bedrock landslide consisting primarily of Eocene Wägital
flysch (Winkler et al., 1985; Schuerch et al., 2006; Golly et al., 2017), while bedrocks in the Lümpenbach and Vogelbach are
largely composed of calcareous sandstones (Milzow et al., 2006). The land cover of the drainage basins consists of alpine
meadows, forests, and wetlands (Turowski et al., 2009; Gies et al., 2022).

**2.2 Sample collection**

From May 2014 to February 2015, surface water was sampled and processed by C. Freymond and H. Gies. Sample
collection from August 2016 to March 2019 was designed to capture variations in OC export in response to both seasonal
changes and shorter-term variations in discharge behavior. We collected surface water samples from the Sihl River (Allmend
Park, 47.35° N, 8.52° E; **Fig. 1a**) in a biweekly rhythm using a river-rinsed bucket. In addition, river water was collected
during 17 storm events, emphasizing discharges >20 m$^3$ s$^{-1}$. Although the water level can rise ~1.7 m during exceptional
flood events, the Sihl River is generally characterized by water depths <1 m suggesting little vertical variations in suspended



sediment and POC concentrations in the water column. Surface waters were retrieved seasonally from Lake Sihl (Sihl River inflow and two locations in the center of the lake) and the Alp River (**Fig. 1a**).

Known volumes of surface water (0.95 to 64.61 L) were filtered through pre-weighed and combusted (450ºC, 6 h) 90 or 142 mm glass microfiber filters (GF/F, Whatman) with a nominal pore size of 0.7 µm using a steel filtration unit. The filtration occurred either in the field or immediately upon returning to the laboratory at ETH. Every sample consists of three identical

replicates. Filters were frozen after filtering and kept frozen until freeze-drying. Dried filters were re-weighed to obtain suspended sediment concentrations (SSC). Filtered water was collected for DOC in 120 mL pre-combusted (450ºC, 6 h) amber bottles, acidified to pH 2 with 85 % $H_3PO_4$ (120 µL), and stored cooled and in the dark. About 4 mL of filtrate was collected in glass vials for the analyses of water isotopic compositions.

**2.3 Geochemical analyses**

Filter pieces (3 mm diameter) containing on average 350 µgC were placed in Ag boats (Säntis Analytical AG) and de-carbonated in a desiccator under HCl vapor (70º C, 72 h), followed by neutralization with NaOH (70º C, 72 h; Freymond et al., 2018). Vapor-acid treated samples were wrapped in tin boats and analyzed for OC content and stable isotopic composition ($\delta^{13}C$) on an elemental analyzer-isotope ratio mass spectrometer (EA-IRMS, Elementar, Vario MICRO cube – Isoprime, VISION) at the Laboratory of Ion Beam Physics (LIP) at ETH Zurich. Radiocarbon ($^{14}C$) was measured directly as

$CO_2$ gas using a mini radiocarbon dating system (MICADAS, Ionplus; Wacker et al., 2010; McIntyre et al., 2016). Samples were calibrated against Oxalic Acid II (NIST SRM 4990C) as well as an in-house soil and shale standards to correct for contamination during fumigation. All samples were corrected for constant contamination (~8 µgC) following Haghipour et al. (2019) and $^{14}C$ data are reported as fraction modern, $F^{14}C$ (Reimer et al., 2004).

A wet chemical oxidation approach was used to convert DOC into $CO_2$ (Lang et al., 2012, 2016). In short, aliquots of DOC

(10-134 µgC) were oxidized using an acidified sodium persulfate solution (100 mL $H_2O$ + 4.0 g $Na_2S_2O_8$ + 200 µL of 85 % $H_3PO_4$) and were purged with high-purity helium gas (Grade 5.0, 99.9999% pure, for 10 min) removing ambient air and inorganic $CO_2$. The samples were heated to 100º C for 1 h to convert the DOC to $CO_2$. The resulting $CO_2$ was analyzed using a MICADAS equipped with a gas-accepting ion source (LIP, ETH Zurich). Blank assessment was based on repeated measurements of sucrose (Sigma, $\delta^{13}C$: −12.4 ‰ VPDB, $F^{14}C$: 1.053±0.003) and phthalic acid (Sigma, $\delta^{13}C$: −33.6 ‰ VPDB,

$F^{14}C$<0.0025) standards. The evaluation of constant contamination amounted to ~2 µgC (Haghipour et al., 2019).

Analysis of water isotopic compositions ($\delta^2H$ and $\delta^{18}O$) was performed on a Picarro L2120-i cavity-ringdown spectrometer (Geological Institute, ETH). Standards comprised VSMOW2, GISP, and SLAP2 as well as three in-house reference waters. Each sample and reference material were injected seven times while discarding the first three injections to eliminate instrumental memory effects.



## 2.4 Bayesian isotope mixing model

MixSIAR is an open-source Bayesian tracer mixing model framework in the R computing environment that allows the estimation of fractional contributions of multiple sources to a mixture (Stock and Semmens, 2016; Stock et al., 2018). The model accounts for uncertainties in the source endmember compositions. Although Bayesian mixing models were originally intended to constrain animal diets in ecology, they are often applied to apportion relative contributions of OC sources in rivers and lakes (Butman et al., 2015; Upadhayay et al., 2017; Repasch et al., 2021). We parameterized the mixing model using POC-$\delta^{13}$C, POC-$F^{14}$C, and three potential endmember compositions assessing the influence of seasonal and hydrodynamic variations on the source apportionment of organic matter in the Sihl River catchment. Organic carbon sources comprised leaf litter, soil (0-40 cm), and bedrock material collected within the Sihl catchment (**Table 2**) (Smith et al., 2013; van der Voort et al., 2016; Gies et al., 2022). MixSIAR was run without any initial assumptions (uninformative prior), a burn-in of 200,000 iterations, a thinning factor of 100, and a chain length of 300,000 for three parallel Markov chain Monte Carlo chains. Model convergence was evaluated using Geweke (Geweke, 1991) and Gelman-Rubin metrics (Gelman and Rubin, 1992).

## 2.5 Estimating fluvial loads using traditional sediment rating curves

Water discharge is a key parameter to discern particulate matter transport in rivers. Commonly, rating curves are used to calculate fluvial export where the sample collection is insufficient to define continuous concentration records. The relationship between concentration (*C*) data and discharge (*Q*) is fitted with a power law function (e.g., Walling, 1977; Cohn, 1995; Syvitski et al., 2000; Wheatcroft et al., 2010), $C=aQ^b\varepsilon$, where the exponent *a* and *b* are rating coefficients inferred from a least linear squares regression of logarithmically transformed data. These rating curves are prone to underestimate long-term sediment transport rates by 10 to 50% (Asselman, 2000; Cohn, 1995; Ferguson, 1986). We apply Duan's (1983) nonparametric retransformation bias correction factor ($\varepsilon$) appropriate for non-normal error distributions to compensate for this underestimation. Non-linear least squares regressions often replace traditional approaches and allow an unbiased estimation of rating curve parameters (Asselman, 2000).

## 2.6 Estimating fluvial loads using machine learning

Although machine learning techniques gain increasing popularity in environmental and earth sciences (e.g., Karpatne et al., 2019; Reichstein et al., 2019), their application in fluvial hydrology currently remains limited (Olyaie et al., 2015; Choubin et al., 2018; Sharafati et al., 2020). Machine (supervised) learning refers to a set of data mining approaches that develop pattern recognition based on a sample dataset in order to predict unlabeled target values. Here, we applied four commonly used machine learning algorithms: a multiple linear regression (MLR), a support vector regression machine (SVR) (Drucker et al., 1997), a random forest regression (RFR) (Breiman, 2001), and a neural network regression (NNR) (McCulloch and



Pitts, 1943). A detailed description of the applied models can be found in **Appendix B**. We evaluated and compared different techniques with the goal of estimating annual sediment and POC fluxes for the Sihl River basin.

A suite of predictor variables was used to construct regression models delineating sediment and OC export rates in the Sihl River. Variables include river discharge (Q), stage (H), precipitation (P), 1-day ($P_{t-1}$), and 2-day ($P_{t-2}$) antecedent precipitation spanning from 1974 to 2020. Daily discharge and water level values were obtained from the gauging station Sihlhölzli, Zurich,

operated by the Swiss Federal Office for the Environment (FOEN, https://www.hydrodaten.admin.ch). Daily precipitation data was retrieved for 21 stations located within and around the Sihl River basin from the Federal Office of Meteorology and Climatology (MeteoSwiss, https://gate.meteoswiss.ch). We applied inverse distance weighted (IDW) interpolation to produce inclusive and comprehensive maps describing the distribution of daily rainfall in the Sihl River watershed. The IDW approach, without considering orography, assumes that the attributed value of an unknown point is the weighted average of known values

within its neighborhood. Weights are inversely related to the distances between the predicted and sampled locations. Input variables a standardized using a robust scaler accounting for skewed data distributions and outliers.

All models were developed using scikit-learn, an open software machine learning library for the Python programming language. We tested several combinations of input predictors (**Table B1**) for machine learning approaches developing regression models. Hyperparameters for MLR, SVR, RFR, and NNR were determined using tuning techniques. Model tuning and stable model

results were derived by 10-fold nested cross-validation (trials=20). The performance of each model to predict suspended sediment and POC concentrations was evaluated based on three commonly used statistical metrics: the coefficient of determination ($R^2$), the root mean squared error (RMSE), and the mean absolute error (MAE). While the $R^2$ indicates the precision of the standard regression type, RMSE and MAE represent the model accuracy. All models were visually examined and compared using a combination of violin and strip plots illustrating the probability and actual distributions of the observed

and predicted data. The best-performing algorithms were chosen to interpolate annual sediment and OC export rates. Predicted sediment and POC concentration values of <0 were set to 0.

### 2.7 Statistical analyses

In order to statistically assess seasonal or rainfall-driven changes in exported sediment and OC concentrations and compositions, we introduce meteorological seasons and the discrimination between baseflow and stormflow conditions as

categorical variables. A discharge threshold value of 12.7 m³ s⁻¹ was derived from the average daily flow duration curve of the Sihl River spanning 47 yr of continuous observation (**Fig. C1**). A flow duration curve represents the frequency of occurrence of various flow rates. Recorded discharges are ranked according to their magnitude and subdivided into the percentages of time during which specific flows are equaled or exceeded. Flow rates ranging from 0 to 10% exceedance are categorized as high flow events, while values above 90% indicate the contributions of groundwater to the streamflow.





Due to the assumption violations of normality, equal variances, and equal sample sizes, we performed non-parametric Mann-Whitney and Kruskal-Wallis rank-sum tests. After the identification of significant between-group differences, we applied Conover-Iman post hoc tests with a Bonferroni adjustment of $p$-values (**Table D1**). All statistical comparisons are reported at the 95 % confidence interval ($p<0.05$).

## 3 Results

### 3.1 Basin hydrology

Mean annual discharges observed during the study period (2016-2019) are comparable to the long-term mean value of $Q_{mean}$ 6.8±0.1 m$^3$ s$^{-1}$ (M±SE). The lowest annual mean discharge is observed in 2018 (6.4±0.4 m$^3$ s$^{-1}$), reflecting prolonged periods of drought (Hari et al., 2020; Peters et al., 2020). The highest annual mean discharge amounts to 7.1±0.4 m$^3$ s$^{-1}$ in 2016. The sampled discharges range from 2.7 to 77 m$^3$ s$^{-1}$ and represent the full range of discharge conditions observed during the 40-

month study period (**Fig. 2a; Fig. C1**). We observe no pronounced seasonal variability in the discharge of the Sihl River. Slight increases in water export coincide with snowmelt and periods of frequent storms in spring and summer (57%). The majority of the annual discharge occurs during storm events (Q>12.7 m$^3$ s$^{-1}$; 82%), while baseflow conditions account for only 18%.

Riverine water isotopic compositions vary from -89.4 to -51.3 ‰ for δ$^2$H values and from -12.7 to -7.7 ‰ for δ$^{18}$O values (**Fig.**

**2c-d**, **3a**; **Table S1**). We note no difference between waters delivered during baseflow and stormflow conditions. However, water isotopic compositions are subject to seasonal shifts. While the majority of precipitation was primarily sourced from the North Atlantic, enriched H and O compositions indicate enhanced moisture supply from terrestrial Mediterranean and locally recycled moisture sources during the summer months (LeGrande and Schmidt, 2006; Batibeniz et al., 2020).

### 3.2 Suspended sediment and organic carbon concentrations

Suspended sediment concentrations range between 0.8 to 133.1 mg L$^{-1}$, with an average of 13.5±2.8 mg L$^{-1}$ (n=77) during low flow conditions (**Fig. 3b**; **Table S1**). The export of SSC reached an observed maximum of 398.3 mg L$^{-1}$ (241.3±28.3 mg L$^{-1}$, n=17) during high discharge events. In comparison, SSC values of the Erlenbach varied from 19.8 to 15,310.7 mg L$^{-1}$ during stormflow (Smith et al., 2013). We observe higher sediment input rates in fall (61.8±21.6 mg L$^{-1}$, n=29) and winter (64.8±25.8mg L$^{-1}$, n=25) compared to the spring (42.1±21.9 mg L$^{-1}$, n=17) and summer (44.1±11.6 mg L$^{-1}$, n=23) months.

Suspended sediments collected from Lake Sihl and the Alp River show concentrations ranging from 1.2 to 82.2 mg L$^{-1}$.

Particulate OC concentrations range from 0.01 to 12.08 mgC L$^{-1}$, with an average concentration of 1.37±0.27 mgC L$^{-1}$ (n=90, **Table S1**). Substantially more organic matter is exported during storm-driven events (5.52±0.81 mgC L$^{-1}$, n=17) than during baseflow conditions (0.41±0.10 mgC L$^{-1}$, n=73). The observed range of suspended sediment OC contents vary from 0.37 to





11.64 wt% (2.48±0.18 wt%, n=92; **Fig. 2e**, **3c**). The mean POC content for low discharges amounts to 2.36±0.17 wt% (n=75),

while POC content rapidly increases (3.02±0.63 wt%, n=17) during storm-driven events. In contrast to SSC, measured OC

contents are lower during the fall (2.33±0.27 wt%, n=29) and winter (2.04±0.18 wt%, n=25) months, and increase to 2.84±0.75

wt% (n =15) in spring and 2.91±0.36 wt% in summer (n=23). The Sihl River transports higher POC contents compared to

values from the Erlenbach reported by Smith et al. (2013; 1.45±0.06 wt%, n=122) and Gies et al. (2022; 1.79±0.34 wt%, n=24).

In contrast, POC contents in the Lümpenbach and Vogelbach of ~5.35 wt% exceed those of the Sihl River (Gies et al., 2022).

Observed POC concentrations and contents vary from 0.04 to 0.76 mg L$^{-1}$ and from 0.89 to 2.13 wt% in Lake Sihl (n=15).

Measured organic matter concentrations in the Alp River average to 0.10±0.07 mg L$^{-1}$ and OC contents to 2.58±0.24 wt%

(n=3).

### 3.3 Isotopic composition of particulate and dissolved organic carbon

Sihl River POC-$\delta^{13}$C signatures across the time-series range from -30.1 to -25.8 ‰ averaging -27.7±0.1 ‰ (n=92; **Fig. 2f**;

**3d**; **Table S1**). No statistically significant differences between POC-$\delta^{13}$C and discharge are observed, but we note

pronounced seasonality (**Table D1**). Higher POC-$\delta^{13}$C values are recorded during the summer (-27.1±0.1 ‰, n=23), whereas

OC measured in spring exhibits on average the lowest $\delta^{13}$C values (-28.1±0.2 ‰, n=17). In contrast, POC-$\delta^{13}$C values in

Lake Sihl (-29.7±0.5 ‰, n=15) and the Alp River (-28.0±0.3 ‰, n=2) are generally lower. Bulk POC-$\delta^{13}$C in the Sihl River

overlaps with reported C$_3$ vegetation and soil biomass constituents (Kohn, 2010; Smith et al., 2013; Gies et al., 2022).

Sihl River POC-F$^{14}$C values range from 0.56 to 1.00 (0.87±0.01, n=91; **Fig. 2g**; **Table S1**), and display a statistically significant

positive correlation with discharge (r$_S$=0.43, $p$<0.001). Mean F$^{14}$C values of 0.86±0.01 (n=75) are measured during low flow

and increase to 0.91±0.01 (n=16; **Fig. 3e**) during high flow conditions, indicating storm-driven mobilization and entrainment

of undegraded, biospheric POC to the Sihl River, the latter having been observed in tectonically active regimes (Lyons et al.,

2002; Carey et al., 2005; Hilton et al., 2008a, 2010; Gomez et al., 2010). In contrast, suspended sediment POC-F$^{14}$C values in

the mountainous Erlenbach average 0.65±0.08 (n=6) (Smith et al., 2013). Gies et al. (2022) report POC-F$^{14}$C signatures for

the Erlenbach, Lümpenbach, and Vogelbach of 0.64±0.22 (n=24), 0.80±0.17 (n=26), and 0.76±0.25 (n=27), respectively. The

depletion in $^{14}$C values in the Erlenbach likely indicates substantial contributions of petrogenic OC. Nevertheless, the overall

OC-F$^{14}$C values of Sihl River POC are in good agreement with forested, temperate catchments characterized by minor inputs

of organic-rich sedimentary bedrock and the absence of intense agricultural land use (Raymond et al., 2004; Longworth et al.,

2007; Goñi et al., 2013). The Alp River and Lake Sihl display POC-F$^{14}$C values of 0.84±0.01 (n=3) and 0.79±0.02 (n=15).

The DOC-F$^{14}$C values vary from 0.52 to 1.16, with a mean of 0.95±0.01 (n=77; **Fig. 2h**, **3f; Table S1**). Moderately aged DOC

is observed in the summer months (0.90±0.03, n=17), whereas DOC enriched in $^{14}$C is discharged during fall (0.97±0.01,

n=23). Similar to the F$^{14}$C signature of POC, high precipitation events supply more modern DOC to the Sihl River (0.98±0.01,

n=17). On average, Lake Sihl (0.97±0.01, n=15) and the Alp River (0.97±0.01, n=2) display slightly higher DOC-F$^{14}$C

signatures than the Sihl River.



### 3.4 Performance of predictive models

All three machine learning algorithms outperform traditional rating curve models (**Fig. 4**; **Fig. B1-2**) in predicting suspended sediment and POC concentrations. The statistical performance of the evaluated models according to different scenarios is listed in **Table B1**. Traditional rating curves overestimate low values of suspended sediment and POC leading to poor

performance. Based on model performance criteria, SSC in the Sihl River depends primarily on discharge, water stage, and 1-day antecedent precipitation as predictor variables. Similar, scenarios that include discharge, water stage, precipitation, and 1-day antecedent precipitation appear to reliably reproduce measured POC concentrations. While discharge and water stage display the highest predictive power for instantaneous SSC, POC concentrations are more accurately described by water stage and 1-day antecedent precipitation. Random forest regression achieves the overall best fit with observed SSC (scenario

7; $R^2$=0.85, RMSE=39.0), followed by SVR (scenario 2; $R^2$=0.81, RMSE=43.8), MLR (scenario 8; $R^2$=0.80, RMSE=45.6), and NNR (scenario 2; $R^2$=0.75, RMSE=48.2). The highest coefficient of determination ($R^2$=0.73) and the lowest root mean squared error (RMSE=1.2) for predicting POC concentrations are obtained from SVR (scenario 4). The performance of NNR (scenario 10; $R^2$=0.70, RMSE=1.4), RFR (scenario 8; $R^2$=0.68, RMSE=1.3), and MLR (scenario 8; $R^2$=0.59, RMSE=1.5) captured observed POC variations with less accuracy.

### 3.5 Annual fluxes and yields of suspended sediment and organic carbon

We calculate suspended sediment and POC export fluxes using continuous 47-year daily water discharge, stage, and precipitation records (see **Sect. 2.6**). Given our intermittent sampling design, we are not able to correct export fluxes for hysteresis effects or supply limitations (Wymore et al., 2019). We regard estimated sediment and POC budgets as conservative estimates, constraining a lower boundary.

Annual suspended sediment flux estimation for the Sihl River range from 17,789.8±1,041.5 (non-linear least squares power law) to 25,788.2±3,775.7 t yr$^{-1}$ (bias-corrected power law) (**Table B2**). Fluxes provided by the best fitting model (RFR) average at 25,166.5±1,055.8 t yr$^{-1}$. The majority of sediment export occurs during storm-driven events (72.9-93.0 %). We observe elevated export fluxes during summer (36.6-48.5 %) and spring (26.4-31.4 %) months corresponding to snowmelt and convective rainfall (Schmidt et al., 2019). The suspended sediment load in fall and winter varies between 12.0 and 18.4 %.

Similar to the export of suspended sediment, modeled annual POC fluxes assume values between 426.3±21.4 (non-linear least squares power law) and 762.7±121.1 t yr$^{-1}$ (bias-corrected power law) (**Table B2**). The mean POC load inferred from the SVR model amounts to 573.5±24.6 t yr$^{-1}$. Particulate organic carbon is primarily mobilized and transported downstream during high discharge events (66.0-94.9 %). The highest POC loads are transported during summer (31.1-49.7 %) and spring (26.2-32.5 %), while lower fluxes are observed in fall and winter (11.5-18.3 %).

The reservoir Lake Sihl is considered a sediment trap, efficiently retaining particulate matter delivered from the upper Sihl River watershed. Therefore, mean annual yield calculations were restricted to the lower Sihl River basin, including the Alp



and Biber catchments. We estimate annual yields between 93.6±5.5 and 135.7±19.9 t km$^{-2}$ yr$^{-1}$ (RFR: 132.4±5.6 t km$^{-2}$ yr$^{-1}$) for suspended sediment and 2.2±0.1 and 4.0±0.6 t km$^{-2}$ yr$^{-1}$ (SVR: 3.0±0.1 t km$^{-2}$ yr$^{-1}$) for POC.

## 4 Discussion

### 310    4.1 Organic carbon source contributions

Rivers integrate a mixture of POC comprising contemporary organic matter supplied by terrestrial and aquatic producers, aged soil-derived organic matter, and OC devoid of $^{14}$C released by weathered sedimentary bedrock (e.g., Hedges et al., 1986; Masiello and Druffel, 2001; Raymond et al., 2004; Blair and Aller, 2012). These sources have distinct carbon isotopic signatures and provide constraints on the contribution from different OC inputs. The Sihl River receives a uniform mixture

of fresh, aged, and ancient OC pools with relatively little variations as seasons progress (**Fig. 5a**). Biospheric carbon sources consist of allochthonous (e.g., vegetation, soils) and autochthonous (phytoplankton, benthic algae, aquatic macrophytes) inputs. Slightly more enriched POC-$\delta^{13}$C values in the summer months may indicate the contribution of freshwater C$_3$ plants ($\delta^{13}$C: ~-18 ‰; Chikaraishi, 2013). However, the coarse-grained riverbed substrate prevents the colonization of macrophytes resulting in poor aquatic vegetation in headwaters and allows only localized growth in the lower reaches of the Sihl River

(Känel et al., 2021). The formation of large-scale phytoplankton blooms and microbial biofilms is likely restricted by the low abundance of nutrients (Känel et al., 2021; Romaní et al., 2004; Battin, 1999) and limited light conditions in forested river segments (Boston and Hill, 1991). Algal growth is further disturbed by high discharge events resulting in river bed movement and the loss of algal mats (Schuwirth et al., 2008). From the above reasoning, we believe that instream biomass does not contribute significant amounts of OC to the Sihl River and would not bias our interpretations.

Soils can often be partitioned into several endmembers reflecting different stages of soil development as aging, microbial decomposition, and respiration introduce alterations to the isotopic composition of organic matter (Fernandez et al., 2003; Werth and Kuzyakov, 2010; Wang et al., 2015). However, Swiss shallow soils display relatively muted gradients in OC-$^{14}$C signatures with increasing soil depth and between climatic regions (van der Voort et al., 2016). This relatively homogenous isotopic composition has been ascribed to the presence of bomb-derived OC in soil layers up to 40 cm depth (van der Voort et

al., 2016, 2019). Van der Voort et al. (2019) suggest that percolation of dissolved organic carbon (as constrained via water-extractable OC measurements) may serve as an agent to propagate modern carbon to deeper soil layers in nonwaterlogged (aerobic) soils, resulting in a less pronounced age gradient with depth. This suggests that physical and chemical soil erosion processes deliver primarily modern dissolved and particulate organic matter to the Sihl River impeding the allocation of distinct sources. Consequently, for the purpose of this study, top (0-20 cm) and deeper (>20 cm) soil horizons are regarded as a single

source.





The legacy of bomb-[14]C is also evident in the DOC fractions retrieved from the Sihl River, which are consistently [14]C-enriched relative to corresponding POC samples (**Fig. 5b**). Commonly, DOC is leached from vegetation and soils by precipitation and its residence time in fluvial systems is similar to that of water (Raymond and Bauer, 2001; Marwick et al., 2015). The overall modern OC-F[14]C signature implies that DOC is primarily sourced from throughfall and the assimilation with non-fossil OC stored in litter and shallow soil layers (Inamdar et al., 2011, 2012). A recent study by von Freyberg et al. (2018) investigated the outflow of Swiss catchments and found that the residence time of contributing groundwater is less than 2-3 months. Similar to the Alp, Biber, Erlenbach, and Vogelbach systems (von Freyberg et al., 2018), the Sihl River water isotopic compositions reflect seasonal cycles in precipitation and streamflow, implying that groundwater contributions are primarily sourced from recent rainfall events (**Fig. 2c-d**). The short residence time and the likely shallow flow paths result in limited fluid and solid interactions impeding the dissolution and mobilization of moderately aged soil organic matter and favoring the export of percolating DOC derived from litter and organic-rich soil horizons.

Aged DOC is often associated with anthropogenic disturbances including deforestation, agriculture (Moore et al., 2013; Drake et al., 2019), atmospheric deposition (Stubbins et al., 2012; Spencer et al., 2014), and the release of petroleum and wastewater (Griffith et al., 2009; Regnier et al., 2013; Butman et al., 2015). Although we observe isolated [14]C-depleted DOC signals collected during the summer months, which could be ascribed to the localized introduction of petrogenic OC emanating from fertilizers, mineral oil, or sewage, the majority of the DOC isotopic compositions indicates an overall low degree of anthropogenic disturbances.

MixSIAR modeling results suggest that suspended sediments in the Sihl River are largely derived from biospheric sources (**Fig. 6**). Fresh plant-derived debris constitutes the primary POC input from winter to spring (48 to 50%), reflecting the considerable extent of forests and grasslands in the Sihl River catchment (**Fig. 1b**). In the summer (68±6%; M±SD) and fall (58±8 %) months, the composition of suspended sediment is dominated by the contribution from eroding soils. This observation is in agreement with soil erosion risk modeling based on the Revised Universal Soil Loss Equation (RUSLE) (Schmidt et al., 2016, 2019). Soil loss peaks between July and September in response to high rainfall erosivity on Swiss grasslands (Schmidt et al., 2019, 2016). Extensive vegetation cover is insufficient to counteract water-driven erosion. Rock-derived OC, abundantly supplied by the highly erodible such as the Eocene Wägital flysch in the Alp watershed, contributes about 12±1 % in fall and spring and 15±2 % in summer and winter. Similar source proportions have been observed in the headwaters of the Alp River (Gies et al., 2022).

**4.2 Downstream evolution of particulate organic carbon**

The isotopic composition of Lake Sihl (open symbols) in the upper watershed is distinctly different from the Sihl River (**Fig. 5a**). While suspended sediment at the inlet of Lake Sihl (Lake Sihl 1) resembles material from the lower Sihl River, suspended sediments within the lake display more depleted [13]C and [14]C signatures. Depleted POC-[13]C signals in lakes can be attributed to enhanced aquatic productivity. The isotopic composition of planktonic freshwater algae can range from -40 to -





22 ‰ with the majority of reported $\delta^{13}C$ being <-28 ‰ (Chikaraishi, 2013 and references therein). Isotopic fractionation of phytoplankton biomass can be amplified in the presence of abundant dissolved inorganic carbon (DIC). Lake Sihl, a

moderately alkaline waterbody, receives DIC from weathering carbonaceous bedrocks (Allochthon nappes, Northern limestone Alps, Säntis zone, **Fig. 1c**) via surface runoff and groundwater inflow. This "hard water effect" further manifests itself in the decrease of POC-F$^{14}$C signatures as the input of bedrock-derived, $^{14}$C-depleted DIC dilutes the carbon isotopic content of the water (Blattmann et al., 2019; Broecker and Walton, 1959; Keaveney and Reimer, 2012). Isotopic shifts in POC may also be caused by the selective uptake, decomposition, and preservation of organic matter (Lehmann et al., 2002,

2004). Kinetic isotope effects during enzymatic reactions lead to the enrichment or depletion of biomolecules relative to the bulk biomass (O'Leary, 1988). Carbohydrates and proteins often enriched in $^{13}$C are bioactive compounds and preferentially decomposed by microbes (Harvey et al., 1995; van Dongen et al., 2002). In contrast, lipids derived from plant tissue or phytoplankton exhibit in general more depleted $\delta^{13}$C values and are more robust against degradation, leading to accumulation in the particulate fraction (Harvey et al., 1995; van Dongen et al., 2002).

Surface water samples collected from Lake Sihl 1 in summer and Lake Sihl 2 in winter and spring are characterized by high SSC, low OC contents, are enriched in $^{13}$C, and depleted in $^{14}$C (**Fig. 5a**). These signatures resemble those of the Erlenbach (Smith et al., 2013; Gies et al., 2022), suggesting enhanced contributions of petrogenic OC (e.g., Wägital flysch). These suspended sediment particles were likely entrained by the Sihl and Minster Rivers in response to storm events, forming sediment plumes in the epilimnion of Lake Sihl (**Fig. 1**). Fine-grained mineral soil- and bedrock-derived particles are advected

to the center of the lake, whereas coarser, waterlogged biospheric debris mobilized by surface runoff is likely deposited near the river inlets (Douglas et al., 2022).

In comparison to the Sihl River, headwater-sourced POC is highly variable and encompasses a large range of carbon isotopic compositions (**Fig. 5a**, **7**). Headwaters, in particular the Erlenbach, receive substantial contributions of petrogenic OC (up to ~40 % of total OC) and fall between modern C$_3$ plants and bedrock endmembers (Smith et al., 2013; Gies et al., 2022). In a

recent study, Hilton et al. (2021) used fluxes of dissolved Re, a redox-sensitive element, to constrain weathering intensities of petrogenic OC in the Erlenbach and Vogelbach basins. Findings suggest that ~40 % of OC contained in the Wägital flysch is lost to oxidative remineralization, allowing the majority of unweathered petrogenic POC to be eroded and entrained into adjacent streams. Despite the high supply of sediment and petrogenic OC, the signal of severely aged organic matter is gradually lost downstream. We attribute the gradual attenuation of headwater OC signals to (1) a declining input and

increasingly distal source of bedrock-derived sediments, (2) an enhanced contribution of modern biospheric OC, (3) abiotic, and (4) biotic processes modifying organic matter during transit.

Highly erodible Eocene flysch sequences are superseded by more competent Cretaceous flysch and molasse units, likely resulting in reduced erosion rates and a lower input of petrogenic OC. Simultaneously, the relative abundance of entrained litter and surface soils increases downstream, thereby diluting or replacing bedrock-derived particles (Feng et al., 2016;

Hemingway et al., 2017). Numerous physicochemical mechanisms dynamically influence the addition, removal, and exchange



of OC in dissolved and particulate pools. These processes include flocculation-deflocculation, particle sorption-desorption, aggregation-disaggregation, leaching, settling, and photo-oxidation (Bauer and Bianchi, 2012; Bianchi and Bauer, 2012). Flocculation and adsorption of largely modern DOC (**Fig. 5b**) onto particles may provide an additional source of biospheric organic matter further masking petrogenic OC inputs (von Wachenfeldt and Tranvik, 2008; Attermeyer et al., 2018). Although

rock-derived carbon is regarded as inert, persisting in the environment for at least millennia, studies have shown that microbes in aquatic settings can assimilate and efficiently respire OC devoid of $^{14}$C to $CO_2$ (Petsch et al., 2001; McCallister et al., 2004; Bouchez et al., 2010). However, flume experiments demonstrate that in-river transport, particle abrasion, and turbulent mixing exert minimal controls on the loss of organic matter and that the preservation of OC is primarily regulated by transient storage in floodplains (Scheingross et al., 2019, 2021). Considering the absence of extensive floodplains and the short transit times in

the Sihl River catchment, we regard oxidative loss as a minor factor contributing to the removal of bedrock-sourced POC (Fox et al., 2020) and assume that overall OC fluxes experience little microbial decomposition during active fluvial transfer.

The downstream exchange and dilution of petrogenic OC with undegraded organic matter have previously been observed in large river systems such as the Amazon (Hedges et al., 1986, 2000; Mayorga et al., 2005), the Ganges-Brahmaputra (Galy et al., 2008; Galy and Eglinton, 2011), the Congo (Hemingway et al., 2017), and Orange rivers (Herrmann et al., 2016). The

alteration of riverine POC composition in these extensive river networks occurs over large spatial scales involving changes in topography, basin morphology, geology, vegetation, and climatic variables. In comparison, the Sihl River integrates and modifies exported POC over a ~40 km river interval without experiencing significant shifts in basin characteristics. These findings imply that low-order rivers may possess the potential to actively transform exported OC impacting local and regional terrestrial carbon cycles.

**4.3 Hydrologic controls on particulate organic carbon sources and pathways**

MixSIAR model results suggest that storm-driven events mobilize and flush enhanced proportions of plant-derived material into the Sihl River, with values rising from 32±9 % during baseflow to 51±12 % during high flow conditions. Concurrently, relative inputs of soil and petrogenic OC decrease from 54±10 % to 39±12 % and from 14±1 % to 10±1 %, respectively.

We observe pronounced patterns in the character of POC isotopic compositions as a function of discharge (**Fig. 7**). During low

flow, POC-$\delta^{13}$C and POC-F$^{14}$C values display a large spread in values, corresponding to heterogeneous contributions from a variety of potential sources. In contrast, the isotopic signatures of storm-derived POC are less variable and appear to converge (as indicated by the arrows in **Fig. 7b-c**) to a POC-$\delta^{13}$C value of -27.5±0.1 ‰ and a POC-F$^{14}$C value of 0.90±0.01. Similar behavior is noted in the Sihl headwaters. Although POC exhibits a larger variance in these headwaters, the composition of OC isotopes forms a relatively homogenous mixture during elevated precipitation events (Gies et al., 2022). This convergence

might indicate a thorough mixing of several carbon pools mobilized during a storm event (Kao and Liu, 2000; Hilton et al., 2008a; Gies et al., 2022). However, higher OC contents and predominately modern $^{14}$C signatures of the Sihl River suspended load point towards a marked shift in sources and transport pathways from moderately aged to fresher organic carbon pools





primarily consisting of surface soils and litter (**Fig. 3 c-d**, **6 a**). The enhanced storm-facilitated export of modern biospheric material has been previously observed in subtropical (Hilton et al., 2008b, 2012; Wang et al., 2016; Qiao et al., 2020) and
temperate regions (Medeiros et al., 2012; Hatten et al., 2012; Goñi et al., 2013), and has been attributed to increased surface runoff and landsliding. Heavy rainfall and the resulting overland flow mobilize and laterally transport loose plant-derived debris and sediment from surface soils to adjacent fluvial systems (Harmon et al., 1986; Medeiros et al., 2012; Hatten et al., 2012; Turowski et al., 2013). Storm-induced erosion processes such as shallow landslides efficiently detach litter and organic-rich surface soil layers and actively connect forested hillslopes to river channels (Hovius et al., 2000; Hilton et al., 2011b,
2012). Storm events may also alter the relative contributions of distal and proximal OC sources. Rising water levels inundate adjacent riparian zones, potentially mobilizing significant amounts of standing riparian biomass, litter, and soil organic matter (Marwick et al., 2014; Sutfin et al., 2016). However, as the vegetation in the Sihl River watershed consists primarily of $C_3$ plants, the distinction between proximal and distal biospheric sources cannot be resolved by a simple carbon isotopic approach.

Best-fit model parameters and predictor variables constraining suspended sediment and POC concentrations show marked
differences indicating a divergence of sources and mobilization pathways as a function of discharge. The traditional rating curve exponent $b$ is often interpreted as a proxy for the mobilization rate of particles in a fluvial system. The slightly higher rating curve exponent for POC ($1.9 \pm 0.1$) compared to SSC ($1.8 \pm 0.1$) indicates that the relative export of organic matter may exceed the export of suspended sediment during elevated discharges. Discharge is the primary descriptive variable predicting SSC in all machine learning techniques followed by water stage and 1-day antecedent precipitation (**Table B1**). In contrast,
the predictor variables water stage and 1-day antecedent precipitation achieve high performance for POC. Similar to the isotopic evidence, enhanced inputs of biospheric OC driven by high precipitation/flooding events may point to increased surface runoff and water erosivity entraining plant- and soil-derived material, while carbon-poor sediment deposited as bank or bedload is likely remobilized within the channel system.

### 4.4 Export fluxes and implications

Sediment fluxes in the Sihl River basin are less than half of the particle export documented in other Swiss Rivers (Spreafico et al., 2005). The low yield can be attributed to (1) the damming of the upper basin (Sihl Lake), retaining annually up to 1470 t yr$^{-1}$ of the suspended sediment load (Spreafico, 2007), (2) fortified banks, (3) topography, and (4) catchment geology. The lithology in the lower watershed consists mainly of molasse characterized by low slope angles and a reduced erosion potential (Schuerch et al., 2006; Korup and Schlunegger, 2009). Only the Alp River drains highly erosive flysch formations
in the lower Sihl River basin (Winkler et al., 1985). However, flysch units differ markedly in their erosivity. While the Erlenbach is underlain by easily erodible, fine-grained Eocene pelitic turbidites and mudstone sequences, bedrock in the Lümpenbach and Vogelbach watersheds mainly consist of more competent Cretaceous calcareous sandstones (**Fig. 1d**) (Keller and Weibel, 1991; Milzow et al., 2006; Schuerch et al., 2006). These differences in lithological units are manifested in their respective sediment yields. The Erlenbach, although comprising 0.4 % of the lower Sihl River basin, supplies about





4.7 to 6.7 % of the overall particulate load, with mean annual sediment yields ranging from 1225 to 1648 t km$^{-2}$ y$^{-1}$ (Keller and Weibel, 1991; Smith et al., 2013). In comparison, the Vogelbach, with a watershed size twice as large as the Erlenbach, displays lower annual sediment yields of 725 t km$^{-2}$ yr$^{-1}$ (Keller and Weibel, 1991) which roughly amounts to similar SSC export flux rates (4.4-6.4 %).

By multiplying OC fluxes with the mean values of the MixSIAR posterior distributions for baseflow and stormflow conditions,
while neglecting contributions from in-situ aquatic productivity, we can estimate export rates for the contributions of vegetation, soil, and bedrock. To further evaluate obtained fluxes and yields, we extract net primary productivity (NPP), soil erosion rates, and soil OC contents for the lower Sihl River basin.

Net primary productivity is defined as the uptake and incorporation of carbon from the atmosphere into plant biomass minus the loss to metabolism and maintenance (Clark et al., 2001). Annual, gap-filled NPP products at a 500 m pixel resolution
(MOD17A3HGFv006, MODIS/Terra; ORNL DAAC 2022) are retrieved and averaged over 20 years. The mean NPP for the lower Sihl River basin amounts to 748±11 tC km$^{-2}$ yr$^{-1}$ which is ~200 tC km$^{-2}$ yr$^{-1}$ higher than the NPP values derived from a Bayesian ecosystem model approach based on forest inventory and flux tower data of *P. abies* (540±150 tC km$^{-2}$ yr$^{-1}$) and *F. sylvatica* (530±100 tC km$^{-2}$ yr$^{-1}$) dominated forests (Trotsiuk et al., 2020). The discrepancy between those two estimates is observed globally and can be attributed to differences in methodology, the classification of the forest cover, and uncertainties
associated with heterogenous respiration, litter, and below-ground biomass (Park et al., 2021). Plant detritus exported as POC in the Sihl River accounts for only a fraction of the carbon sequestered and stored in vegetation (0.13 to 0.38 %) and is in good agreement with global estimates of fluvially exported terrestrial biospheric OC (Galy et al., 2015).

We extract and summarize monthly soil loss rates Schmidt et al. (2019) (82.8±12.1 t km$^{-2}$ yr$^{-1}$), while average soil OC contents (0-120 cm, 1.7±0.6 wt%) are derived from Nussbaum et al. (2014) and Solly et al. (2020). The resulting soil OC loss of 1.4±0.5
tC km$^{-2}$ yr$^{-1}$ is consistent with the computed Sihl River soil export yield ranging from 1.0 to 1.6 tC km$^{-2}$ yr$^{-1}$. Similar soil OC erosion rates have been estimated for the Lümpenbach (0.7±0.3 tC km$^{-2}$ yr$^{-1}$) and Vogelbach (0.8±0.4 tC km$^{-2}$ yr$^{-1}$) watersheds (Gies et al., 2022). In contrast, the Erlenbach displays reduced soil OC yields of 0.3±0.2 tC km$^{-2}$ yr$^{-1}$. The geomorphology in the Erlenbach watershed rather favors the export of bedrock material due to abrasion and creep landslides (Schuerch et al., 2006; Smith et al., 2013).

Biospheric (2.0-3.6 tC km$^{-2}$ yr$^{-1}$) and petrogenic (0.3-0.4 tC km$^{-2}$ yr$^{-1}$) POC yields in the Sihl River are similar to contributions from the Erlenbach reported by Gies et al. (2022) (POC$_{bio}$: 1.2±0.4 tC km$^{-2}$ yr$^{-1}$) but distinctly lower than estimations by Smith et al. (2013) (POC$_{bio}$: 14.0±4.4 tC km$^{-2}$ yr$^{-1}$, POC$_{petro}$: 10.1±1.6 tC km$^{-2}$ yr$^{-1}$). Smith et al. (2013) focused their sample collection on storm-driven events yielding high export fluxes. However, the increase in relative biospheric contributions concomitant with the reduction in petrogenic OC proportions, and the decline in absolute export
yields from headwaters to the downstream Sihl sampling site, underline the impact of diverse processes acting upon and contributing to OC along the fluvial continuum. These processes reflect the impact of subtle changes in basin lithology





(erosivity, OC content), geomorphology (slope, floodplain extent), and anthropogenic activities (e.g., damming, channelization, land-use) on the age and the composition of exported POC.

**5 Conclusions**

This study focuses on temporal variations in organic carbon export from a Swiss subalpine river and provides insights into the mechanisms of sediment and associated OC mobilization and transport within a moderately steep river basin. Our results indicate that POC in the Sihl River consists primarily of modern to moderately aged biospheric OC derived from terrestrial vegetation and soils. In summer, severe storm events promote water erosivity and the entrainment of soil-sourced particles. While petrogenic carbon is prevalent in Sihl headwater catchments, the signal is gradually lost downstream. We associate this

decline of rock-derived OC with decreasing contributions of source material that are restricted to upstream segments of the watershed, the dilution and replacement by soil and plant biomass, and instream OC transformation processes. Despite the low stream gradient of the Sihl River, particle export is driven by episodic, short-lived storm events. We observe large variations in the isotopic composition of organic matter during baseflow conditions, whereas POC-$\delta^{13}$C and POC-F$^{14}$C values converge to a more uniform mixture during storm-driven events. Results of traditional and machine learning modeling approach further

reveal diverging transport pathways for suspended sediment and OC with increasing discharge. Given the high POC content, the modern POC-F$^{14}$C signature, and the differences in particle mobilization, we suggest that severe precipitation events facilitate the preferential entrainment of litter and surficial soil layers via surface runoff and shallow landsliding or via the increased input from proximal riparian vegetation and soils. Climate model simulations predict an increase in intensity, frequency, and duration of extreme precipitation events both regionally and globally with global warming (Myhre et al., 2019;

Kahraman et al., 2021) resulting in enhanced flood risks, water-induced erosion, and landsliding. The increased export of freshwater, nutrients, and sediment will likely severely affect downstream ecosystems, carbon cycling, requiring direct human intervention (Turowski et al., 2009; Talbot et al., 2018), and warranting the continuous monitoring of river systems.

**Appendix A: Dissolved organic carbon concentrations**

DOC concentration measurements were conducted using a Shimadzu system (TOC-L Series) at the Department of

Environmental System Science at ETH Zurich and are reported in **Table S1**. However, measurements from August 2016 to March 2018 are not reported due to uncertainties in the quality of the measurements and we choose not to discuss them in the manuscript.

**Appendix B: Machine learning approaches**

Multiple linear regression (MLR) assumes a linear relationship between a single dependent continuous variable and several independent variables. To reduce overfitting of the MLR, we apply Elastic Net regularization which penalizes the model for both $\ell_1$ and $\ell_2$-norms (Zou and Hastie, 2005).

Support vector regressions (SVR) are a popular machine learning approach performing linear or nonlinear classification, regression, and outlier detection. The support vector machine classification algorithm identifies an optimal hyperplane in n-dimensional space to separate and categorize data points. In contrast, the SVR uses this principle to fit as many instances onto 530 a hyperplane while limiting margin violations. This supervised learning algorithm supports different kernels (linear, gaussian radial basis, polynomial) handling nonlinearity. Radial basis function kernels are commonly applied to fit non-linear regression lines and often outperform linear and polynomial kernels.

A standard decision tree is a non-parametric supervised learning method that predicts the value of a target variable by inferring simple decision rules based on data features. Decision trees are prone to overfitting the training set and thus are often replaced 535 by an ensemble of decision trees called a random forest. Random forests are generally built on bagging and random feature selection creating an uncorrelated forest of decision trees and thus generalizing well to unseen data (Breiman, 2001).

Neural Networks are a system of algorithms inspired by the human brain that attempts to recognize underlying patterns in a data set. The simplest neural network consists of an input and output layer that is interconnected through one hidden layer. Each neuron in these layers has an associated weight and threshold. The weighted sum of all neurons in a layer is passed 540 through an activation function and augmented by a bias term. In a feed-forward neural network, backpropagation adjusts the weights by minimizing the loss function and reducing the error between modeled and output values (Rumelhart et al., 1986). The utilized architecture consists of two hidden layers containing each ten neurons. To prevent overfitting, we apply dropout. This regularization technique temporarily removes units during the training period (Srivastava et al., 2014). Dropped units are chosen randomly.

**Appendix C: Flow duration curve**

**Appendix D: Results of non-parametric analyses of variance**

**Data availability**

All data generated are submitted and will be openly available in the EarthChem Library.



**Supplement**

Supporting information is available for this paper

**Author contributions**

MSS, ML, and TIE led the design of the study. HG and CF led data collection and analyses from 2014 to 2015. MSS conducted field and lab work from 2016 to 2019, data analysis and interpretation. NH contributed to laboratory analyses. MSS prepared the manuscript with contributions from all co-authors.

**Competing interests**

The authors declare that they have no conflict of interest.

**Acknowledgments**

We thank Francien Peterse for her motivation to initiate time-series sampling of the Sihl River. We thank Lena Märki for sample collection and Daniel Montluçon for laboratory assistance.

**Financial support**

M.S.S. was supported by the Swiss National Science Foundation through the grants SNF200020_163162/1, "CAPS-LOCK II" and SNF200020_184865/1, "CAPS-LOCK III".

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



**Table 1:** Summary of river and watershed characteristics for the Sihl River, Erlenbach, Lümpenbach, and Vogelbach. Information for the Sihl River headwater streams is provided by Smith et al. (2013) and von Freyberg et al. (2018).

| | | Erlenbach | Lümpenbach | Vogelbach | Sihl - Sihlhölzli |
|---|---|---|---|---|---|
| Basin size at gauging station (km$^2$)[a] | | 0.73 | 0.88 | 1.58 | 346.02 |
| Average slope (°)[a] | | 23.9 | 19.6 | 28.9 | 19.5 |
| Mean catchment elevation (m a.s.l.)[a] | | 1359 | 1336 | 1335 | 1041 |
| Average discharge (m$^3$ s$^{-1}$)[a] | | 0.04 | 0.05 | 0.07 | 6.83 |
| Basin geology[b] | | Eocene flysch, Cretaceous flysch | Cretaceous flysch | Cretaceous flysch | Subalpine molasse, flysch, limestone, evaporites, sandstone |
| Land-use[c] | Settlements (%) | 0 | 0 | 0 | 9.0 |
| | Agriculture (%) | 0 | 0 | 0 | 1.1 |
| | Meadows, pastures (%) | 21.6 | 81 | 30 | 38.9 |
| | Forest (%) | 59.5 | 19 | 70 | 43.4 |
| | Water bodies (%) | 0 | 0 | 0 | 1.3 |
| | Unproductive areas (%) | 18.9 | 0 | 0 | 6.1 |

[a] Federal Office for the Environment, https://www.bafu.admin.ch/bafu/de/home.html

[b] Federal Office of Topography swisstopo, https://www.swisstopo.admin.ch/

[c] Federal Statistic Office, https://www.bfs.admin.ch/bfs/en/home.html


**Table 2:** Organic carbon endmember compositions used in the MixSIAR Bayesian model.

| | POC-$\delta^{13}$C (‰) | | | POC-F$^{14}$C | | | Reference |
|---|---|---|---|---|---|---|---|
| | n | M | SD | n | M | SD | |
| Bedrock | 22 | -25.71 | 0.84 | est. | 0.00 | 0.01 | Smith et al. (2013), Gies et al. (2022) |
| Soil | 33 | -27.20 | 0.76 | 50 | 1.00 | 0.10 | Smith et al. (2013), van der Voort et al. (2016), Gies et al. (2022) |
| Vegetation | 21 | -27.37 | 1.62 | 1 | 1.02 | 0.01 | Smith et al. (2013), Gies et al. (2022) |





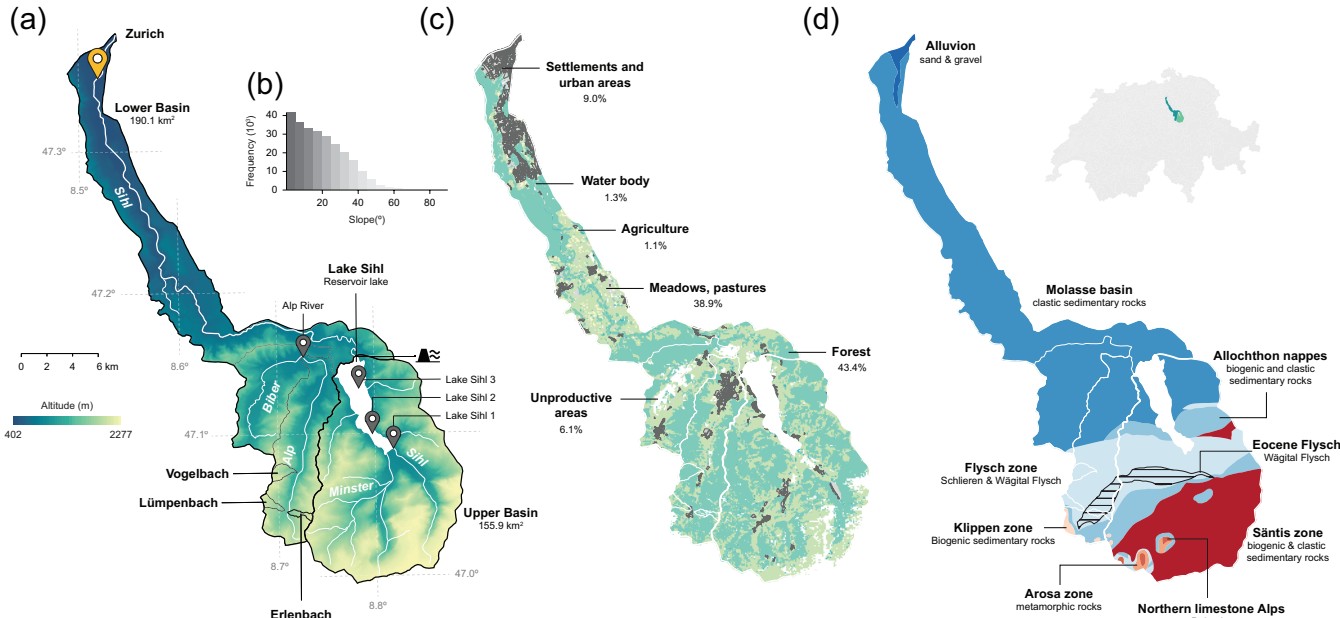

**Figure 1:** Sihl River basin showing (a) altitude, (b) the distribution of slope angles (https:www.swisstopo.admin.ch), (c) different land-use
types (https://www.bfs.admin.ch), and (d) the underlying geology (https:www.swisstopo.admin.ch, Winkler et al., 1985). The sampling
location for the Sihl River time series is indicated as a yellow symbol; the locations of the Lake Sihl and the Alp River sampling sites are
shown as grey symbols.









**Figure 2:** Hydrographs for the sampling periods from May 2014 to February 2015 and August 2016 to March 2019: (a) hourly discharge values (m³ s⁻¹; https://www.hydrodaten.admin.ch) and (b) daily precipitation values for the Sihl River basin (mm; https://gate.meteoswiss.ch). Gray dots represent individual sampling campaigns. Water isotopic compositions, (c) $\delta^2H$ (‰) and (d) $\delta^{18}O$ (‰), are shown alongside (e) particulate organic carbon (POC) contents (wt%), (f) POC-$\delta^{13}C$ (‰,), (g) POC-F$^{14}$C, and (h) dissolved organic carbon (DOC) F$^{14}$C. Dots are scaled to discharge.







**Figure 3:** Combined violin and strip plots of (a) $\delta^{18}$O (‰), (b) suspended sediment concentrations (SSC, mg L$^{-1}$), (c) particulate organic carbon (POC) content (wt%), (d) POC-$\delta^{13}$C (‰), (e) POC-F$^{14}$C, and (f) dissolved organic carbon (DOC) F$^{14}$C faceted for seasons and discharge. Violin plots depict rotated kernel density plots. White vertical lines indicate median values. Significant between-group differences are denoted with brackets and *p*-values (**Table D1**).

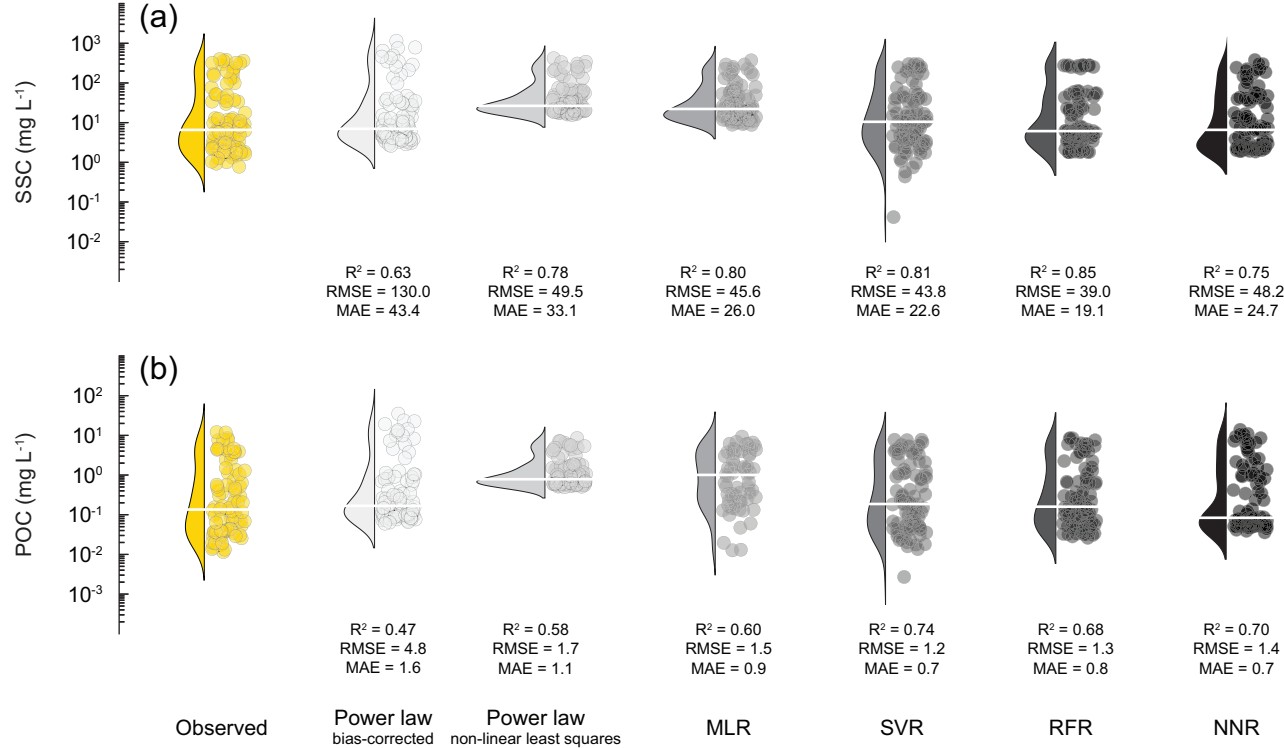


**Figure 4:** Combined violin and strip plots comparing model performances in predicting (a) suspended sediment (SSC, mg L$^{-1}$) and (b) particulate organic carbon (POC, mg L$^{-1}$) concentrations. Models approaches include traditional and non-linear least squares power law functions as well as multilinear regression (MLR), support vector regression (SVR), random forest regression (RFR), and neural network regression (NNR). Violin plots depict rotated kernel density plots, with white horizontal lines indicating median values.




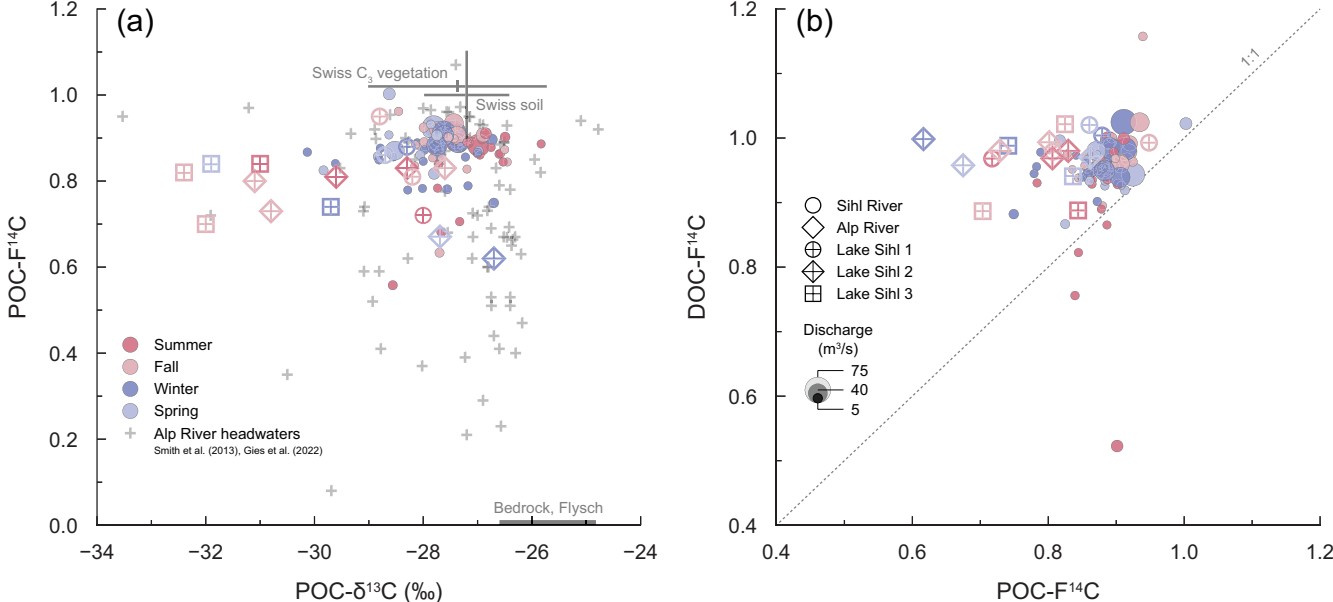

**Figure 5:** Relationship between (a) particulate organic carbon (POC) F[14]C and POC-δ[13]C (‰) values and (b) between dissolved organic carbon (DOC) and POC-F[14]C values. Gray crosses indicate samples from the Alptal headwater streams: Erlenbach, Lümpenbach, and Vogelbach (Smith et al., 2013; Gies et al., 2022). Sources for plant, soil, and bedrock endmembers are listed in Table 2. Dots are color-coded for seasons and scaled to discharge.

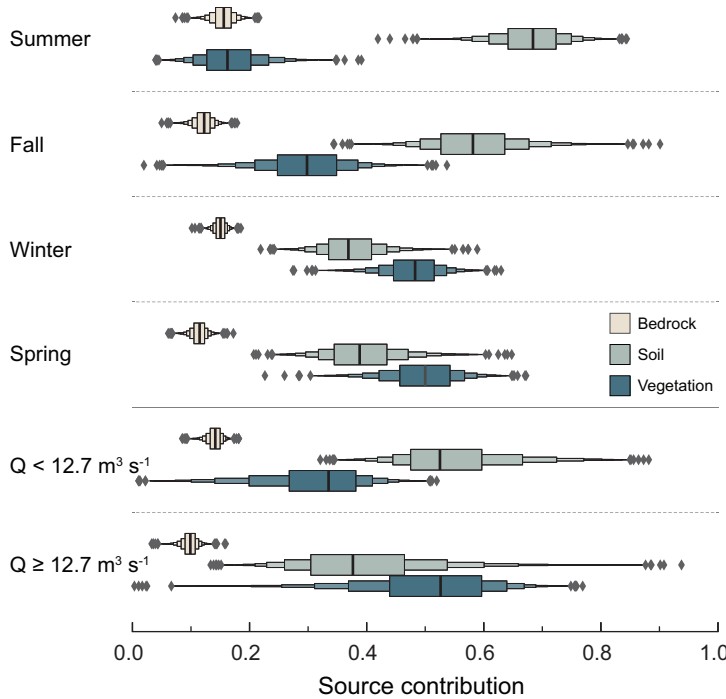



**Figure 6:** Boxenplots depicting MixSIAR-derived posterior distributions of bedrock, soil, and terrestrial vegetation endmember contributions to the bulk POC load according to seasons and changes in discharge. The innermost box is drawn at the lower and upper quartiles. Incrementally narrower boxes represent the lower and upper octiles, hexadeciles, and so forth. Outliers are denoted with diamonds, while black lines depict medians.






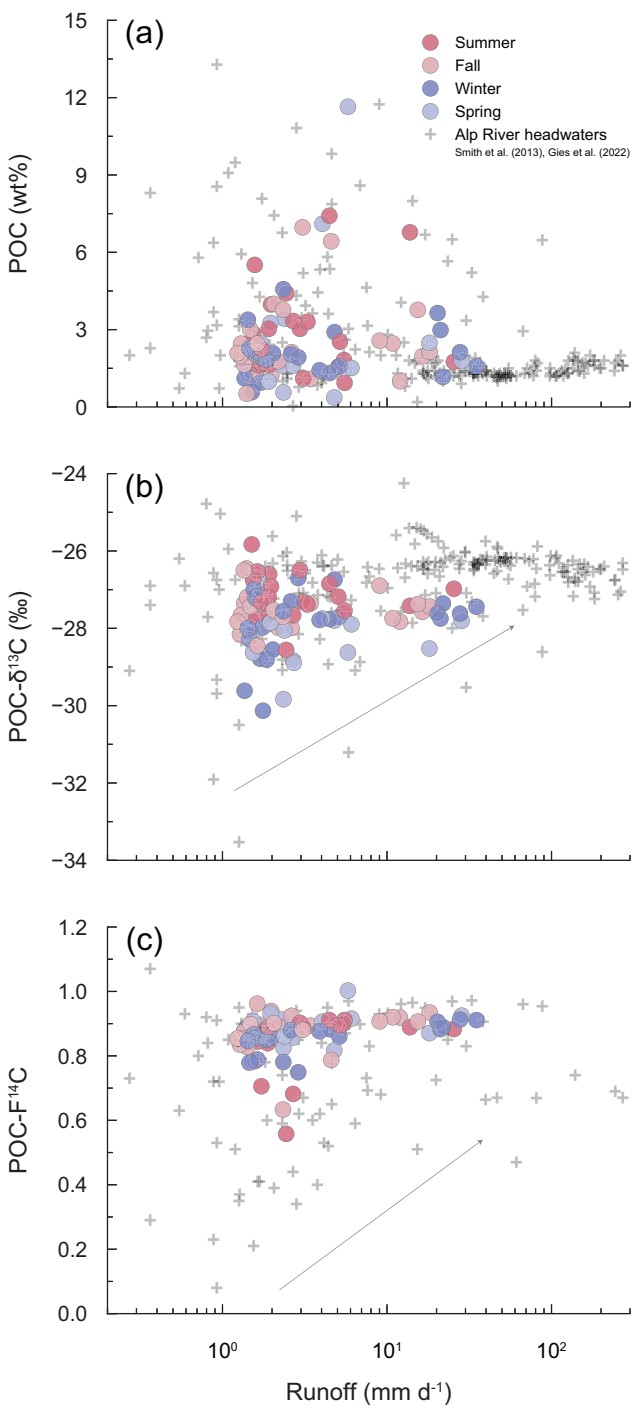

**Figure 7:** Relationship between runoff (mm d⁻¹), (a) particulate organic carbon (POC) content (wt%), (b) POC-$\delta^{13}$C (‰), and (c) POC-F$^{14}$C. Gray crosses indicate samples from the Erlenbach, Lümpenbach, and Vogelbach (Gies et al., 2022; Smith et al., 2013). Circles are color-coded for seasons.



**Table B1:** Model performance in predicting suspended sediment and particulate organic carbon concentrations for the investigated period.

| Model | Scenario | Model structure | $R^2$ | RMSE (mg L$^{-1}$) | MAE (mg L$^{-1}$) | | $R^2$ | RMSE (mg L$^{-1}$) | MAE (mg L$^{-1}$) |
|---|---|---|---|---|---|---|---|---|---|
| **Suspended Sediment Concentration** | | | | | | | | | |
| *Power Law (biased corrected)* | | | | | | *Power Law (non-linear least squares)* | | | |
| | | | 0.628 | 130.027 | 43.355 | | 0.777 | 49.534 | 33.086 |
| *Multiple Linear Regression (Elastic Net)* | | | | | | *Random Forest Regression* | | | |
| | 1 | SSC ~ Q | 0.769 | 48.241 | 26.720 | | 0.831 | 41.617 | 20.951 |
| | 2 | SSC ~ Q + H | 0.781 | 47.119 | 26.052 | | 0.789 | 47.633 | 24.752 |
| | 3 | SSC ~ Q + H + P | 0.775 | 47.877 | 26.440 | | 0.809 | 46.130 | 23.236 |
| | 4 | SSC ~ Q + H + P + $P_{t-1}$ | 0.791 | 46.171 | 26.236 | | 0.829 | 41.131 | 20.961 |
| | 5 | SSC ~ Q + H + P + $P_{t-1}$ + $P_{t-2}$ | 0.791 | 46.165 | 26.597 | | 0.823 | 44.162 | 21.384 |
| | 6 | SSC ~ $P_{t-1}$ | 0.530 | 66.768 | 40.194 | | 0.589 | 67.226 | 28.772 |
| | 7 | SSC ~ $P_{t-1}$ + Q | 0.792 | 45.974 | **25.824** | | **0.847** | **39.027** | **19.136** |
| | 8 | SSC ~ $P_{t-1}$ + Q + H | **0.796** | **45.568** | 26.007 | | 0.783 | 41.032 | 20.784 |
| | 9 | SSC ~ H | 0.673 | 57.444 | 34.489 | | 0.670 | 56.707 | 32.647 |
| | 10 | SSC ~ H + $P_{t-1}$ | 0.666 | 57.614 | 35.160 | | 0.659 | 54.453 | 27.061 |
| | 11 | SSC ~ P | 0.335 | 81.021 | 56.256 | | 0.115 | 99.313 | 58.731 |
| *Support Vector Regression* | | | | | | *Neural Network Regression* | | | |
| | 1 | SSC ~ Q | 0.796 | 45.110 | 23.504 | | 0.717 | 50.760 | 27.259 |
| | 2 | SSC ~ Q + H | **0.810** | **43.846** | 22.647 | | **0.752** | **48.235** | 24.726 |
| | 3 | SSC ~ Q + H + P | 0.751 | 50.146 | 25.362 | | 0.746 | 48.390 | 25.518 |
| | 4 | SSC ~ Q + H + P + $P_{t-1}$ | 0.771 | 48.357 | 22.429 | | 0.728 | 48.316 | **24.640** |
| | 5 | SSC ~ Q + H + P + $P_{t-1}$ + $P_{t-2}$ | 0.763 | 48.937 | 23.132 | | 0.734 | 48.845 | 25.634 |
| | 6 | SSC ~ $P_{t-1}$ | 0.358 | 79.794 | 37.551 | | 0.435 | 71.182 | 32.648 |
| | 7 | SSC ~ $P_{t-1}$ + Q | 0.793 | 45.320 | 21.168 | | 0.690 | 49.812 | 25.873 |
| | 8 | SSC ~ $P_{t-1}$ + Q + H | 0.799 | 44.745 | **20.701** | | 0.741 | 49.391 | 25.241 |
| | 9 | SSC ~ H | 0.479 | 70.756 | 34.971 | | 0.593 | 59.417 | 28.658 |
| | 10 | SSC ~ H + $P_{t-1}$ | 0.604 | 62.900 | 29.319 | | 0.669 | 56.086 | 27.110 |
| | 11 | SSC ~ P | 0.168 | 91.156 | 47.051 | | 0.023 | 97.894 | 52.224 |
| **Particulate Organic Carbon Concentration** | | | | | | | | | |
| *Power Law (biased corrected)* | | | | | | *Power Law (non-linear least squares)* | | | |
| | | | 0.474 | 4.755 | 1.568 | | 0.584 | 1.678 | 1.069 |
| *Multiple Linear Regression (Elastic Net)* | | | | | | *Random Forest Regression* | | | |
| | 1 | SSC ~ Q | 0.362 | 1.780 | 1.079 | | 0.467 | 1.569 | 0.874 |
| | 2 | SSC ~ Q + H | 0.555 | 1.506 | 0.841 | | 0.663 | 1.383 | 0.765 |
| | 3 | SSC ~ Q + H + P | 0.440 | 1.676 | 0.974 | | 0.613 | 1.356 | 0.780 |
| | 4 | SSC ~ Q + H + P + $P_{t-1}$ | 0.592 | 1.499 | 0.884 | | 0.642 | 1.381 | 0.784 |
| | 5 | SSC ~ Q + H + P + $P_{t-1}$ + $P_{t-2}$ | 0.578 | 1.526 | 0.913 | | 0.601 | 1.518 | 0.847 |
| | 6 | SSC ~ $P_{t-1}$ | 0.368 | 1.757 | 1.197 | | 0.628 | 1.598 | 0.781 |
| | 7 | SSC ~ $P_{t-1}$ + Q | 0.464 | 1.660 | 0.987 | | 0.503 | 1.487 | 0.866 |
| | 8 | SSC ~ $P_{t-1}$ + Q + H | **0.595** | **1.485** | 0.871 | | **0.680** | **1.327** | **0.749** |
| | 9 | SSC ~ H | 0.553 | 1.509 | **0.839** | | 0.479 | 1.522 | 0.864 |
| | 10 | SSC ~ H + $P_{t-1}$ | 0.515 | 1.601 | 0.999 | | 0.638 | 1.503 | 0.747 |
| | 11 | SSC ~ P | 0.078 | 2.006 | 1.452 | | -0.263 | 2.218 | 1.534 |
| *Support Vector Regression* | | | | | | *Neural Network Regression* | | | |
| | 1 | SSC ~ Q | 0.064 | 2.061 | 1.144 | | 0.627 | 1.471 | 0.792 |
| | 2 | SSC ~ Q + H | 0.636 | 1.422 | 0.791 | | 0.579 | 1.599 | 0.817 |
| | 3 | SSC ~ Q + H + P | 0.663 | 1.343 | 0.729 | | 0.574 | 1.411 | 0.760 |
| | 4 | SSC ~ Q + H + P + $P_{t-1}$ | **0.735** | **1.226** | **0.670** | | 0.650 | 1.479 | 0.743 |
| | 5 | SSC ~ Q + H + P + $P_{t-1}$ + $P_{t-2}$ | 0.594 | 1.531 | 0.862 | | 0.285 | 1.762 | 0.840 |
| | 6 | SSC ~ $P_{t-1}$ | 0.498 | 1.648 | 0.848 | | 0.590 | 1.699 | 0.887 |
| | 7 | SSC ~ $P_{t-1}$ + Q | 0.531 | 1.541 | 0.797 | | 0.667 | 1.432 | 0.779 |
| | 8 | SSC ~ $P_{t-1}$ + Q + H | 0.612 | 1.455 | 0.762 | | 0.656 | 1.468 | 0.781 |
| | 9 | SSC ~ H | 0.541 | 1.579 | 0.748 | | 0.599 | 1.466 | 0.713 |
| | 10 | SSC ~ H + $P_{t-1}$ | 0.471 | 1.598 | 0.792 | | **0.697** | **1.392** | **0.701** |
| | 11 | SSC ~ P | -0.022 | 2.305 | 1.245 | | 0.188 | 2.250 | 1.168 |





**Table B2:** Modelled export of suspended sediment and particulate organic carbon using traditional and machine learning approaches, averaged over 47 yr (1974-2020 inclusive).

| | | Mean annual flux (t) | | | | | |
|---|---|---|---|---|---|---|---|
| | | Q < 12.7 m s⁻¹ | Q > 12.7 m s⁻¹ | Summer | Fall | Winter | Spring |
| **Suspended Sediment** | | | | | | | |
| Power Law (bias-corrected) | 25,788.18±3,775.68 | 1,816.35±48.17 (7.04 %) | 23,971.83±3,765.95 (92.96 %) | 12,498.98±3,055.68 (48.47 %) | 3,373.31±615.71 (13.08 %) | 3,100.66±500.32 (12.02 %) | 6,815.23±2,343.25 (26.43 %) |
| Power Law (non-linear least squares) | 17,789.77±1,041.51 | 4,821.24±91.88 (27.10 %) | 12,968.53±1,009.22 (72.90 %) | 6,502.93±748.95 (36.55 %) | 3,061.53±357.74 (17.21 %) | 3,023.15±295.25 (16.99 %) | 5,202.16±716.82 (29.24 %) |
| Multiple Linear Regression (Elastic Net) | 25,455.62±1,595.16 | 5,885.17±146.12 (23.12 %) | 19,570.46±1,534.49 (76.88 %) | 9,550.15±1,140.49 (37.52 %) | 4,241.03±551.88 (16.66 %) | 4,174.50±453.24 (16.40 %) | 7,489.94±1,092.33 (29.42 %) |
| Support Vector Regression | 19,673.18±842.56 | 4,055.42±127.67 (20.61 %) | 15,617.76±763.59 (79.39 %) | 6,337.29±535.88 (32.21 %) | 3,557.99±3,557.99 (18.09 %) | 3,603.13±362.26 (18.31 %) | 6,174.77±562.59 (31.39 %) |
| Random Forest Regression | 25,166.54±1,055.81 | 5,675.07±193.27 (22.55 %) | 19,491.48±931.47 (77.45 %) | 8,151.50±627.60 (32.39 %) | 4,618.15±522.81 (18.35 %) | 4,514.11±431.31 (17.94 %) | 7,882.78±752.69 (31.32 %) |
| Neural Network Regression | 24,854.07±1,741.07 | 3,841.13±118.92 (15.45 %) | 21,012.94±1,689.47 (84.55 %) | 9,651.12±1,260.01 (38.83 %) | 4,013.30±576.11 (16.15 %) | 3,900.18±478.11 (15.69 %) | 7,289.47±1,194.11 (29.33 %) |
| **Particulate Organic Carbon** | | | | | | | |
| Power Law (bias-corrected) | 762.68±121.06 | 45.47±1.24 (5.96 %) | 717.21±120.82 (94.04 %) | 379.06±98.69 (49.70 %) | 96.21±18.21 (12.61 %) | 87.57±14.70 (11.48 %) | 199.84±74.24 (26.20 %) |
| Power Law (non-linear least squares) | 426.30±21.39 | 136.13±2.42 (31.93 %) | 290.17±20.46 (68.07 %) | 148.80±14.94 (34.90 %) | 75.80±8.07 (17.78 %) | 75.42±6.63 (17.69 %) | 126.29±14.84 (29.62 %) |
| Multiple Linear Regression (Elastic Net) | 638.48±33.50 | 200.05±5.52 (31.33 %) | 438.44±30.37 (68.67 %) | 227.97±22.17 (35.71 %) | 109.50±13.67 (17.15 %) | 108.35±10.94 (16.97 %) | 192.66±22.42 (30.17 %) |
| Support Vector Regression | 573.48±24.59 | 156.04±4.89 (27.21 %) | 417.44±21.36 (72.79 %) | 187.97±14.85 (32.78 %) | 104.64±12.58 (18.25 %) | 100.16±9.29 (17.46 %) | 180.70±17.00 (31.51%) |
| Random Forest Regression | 539.98±21.56 | 183.60±5.70 (34.00 %) | 356.38±17.61 (66.00 %) | 167.54±12.25 (31.10 %) | 98.29±10.62 (18.20 %) | 98.39±8.60 (18.22 %) | 175.36±15.03 (32.48%) |
| Neural Network Regression | 580.23±32.64 | 136.21±4.30 (23.48 %) | 444.01±30.25 (76.52 %) | 209.88±22.26 (36.17 %) | 97.41±12.92 (16.79 %) | 95.15±10.37 (16.40 %) | 177.79±22.19 (30.64%) |


**Table D1:** Results of significant non-parametric Mann-Whitney and Kruskal-Wallis ranks sum tests as well as Conover-Iman post hoc tests.

| | Mann-Whitney rank sum test | | | Kruskal-Wallis rank sum test | | | | Conover-Iman post hoc test | | |
|---|---|---|---|---|---|---|---|---|---|---|
| | | difference | *p*-value | | df | H | *p*-value | | difference | *p*-value |
| **SSC (mg/L)** | Baseflow - stormflow | -231.64 | **<0.001** | δ²H (‰) | 3 | 25.98 | **<0.001** | Fall - Spring | 4.23 | **<0.001** |
| | | | | | | | | Fall - Summer | -0.81 | 1.000 |
| **POC-F¹⁴C** | Baseflow - stormflow | -0.04 | **<0.001** | | | | | Spring - Summer | -4.89 | **<0.001** |
| | | | | | | | | Fall - Winter | 3.86 | **0.001** |
| **DOC-F¹⁴C** | Baseflow - stormflow | -0.03 | **<0.001** | | | | | Spring - Winter | -0.54 | 1.000 |
| | | | | | | | | Summer - Winter | 4.56 | **<0.001** |
| | | | | δ¹⁸O (‰) | 3 | 24.25 | **<0.001** | Fall - Spring | 4.29 | **<0.001** |
| | | | | | | | | Fall - Summer | -0.74 | 1.000 |
| | | | | | | | | Spring - Summer | -4.88 | **<0.001** |
| | | | | | | | | Fall - Winter | 3.35 | **0.004** |
| | | | | | | | | Spring - Winter | -1.08 | 0.853 |
| | | | | | | | | Summer - Winter | 3.99 | **<0.001** |
| | | | | POC-δ¹³C (‰) | 3 | 24.05 | **<0.001** | Fall - Spring | 2.73 | **0.023** |
| | | | | | | | | Fall - Summer | -3.03 | **0.010** |
| | | | | | | | | Spring - Summer | -5.33 | **<0.001** |
| | | | | | | | | Fall - Winter | 1.15 | 0.761 |
| | | | | | | | | Spring - Winter | -1.67 | 0.294 |
| | | | | | | | | Summer - Winter | 4.08 | **<0.001** |
| | | | | DOC-F¹⁴C | 3 | 10.55 | **0.010** | Fall - Spring | 2.20 | 0.092 |
| | | | | | | | | Fall - Summer | 3.29 | **0.005** |
| | | | | | | | | Spring - Summer | 0.91 | 1.000 |
| | | | | | | | | Fall - Winter | 1.34 | 0.556 |
| | | | | | | | | Spring - Winter | -0.99 | 0.970 |
| | | | | | | | | Summer - Winter | -2.03 | 0.139 |



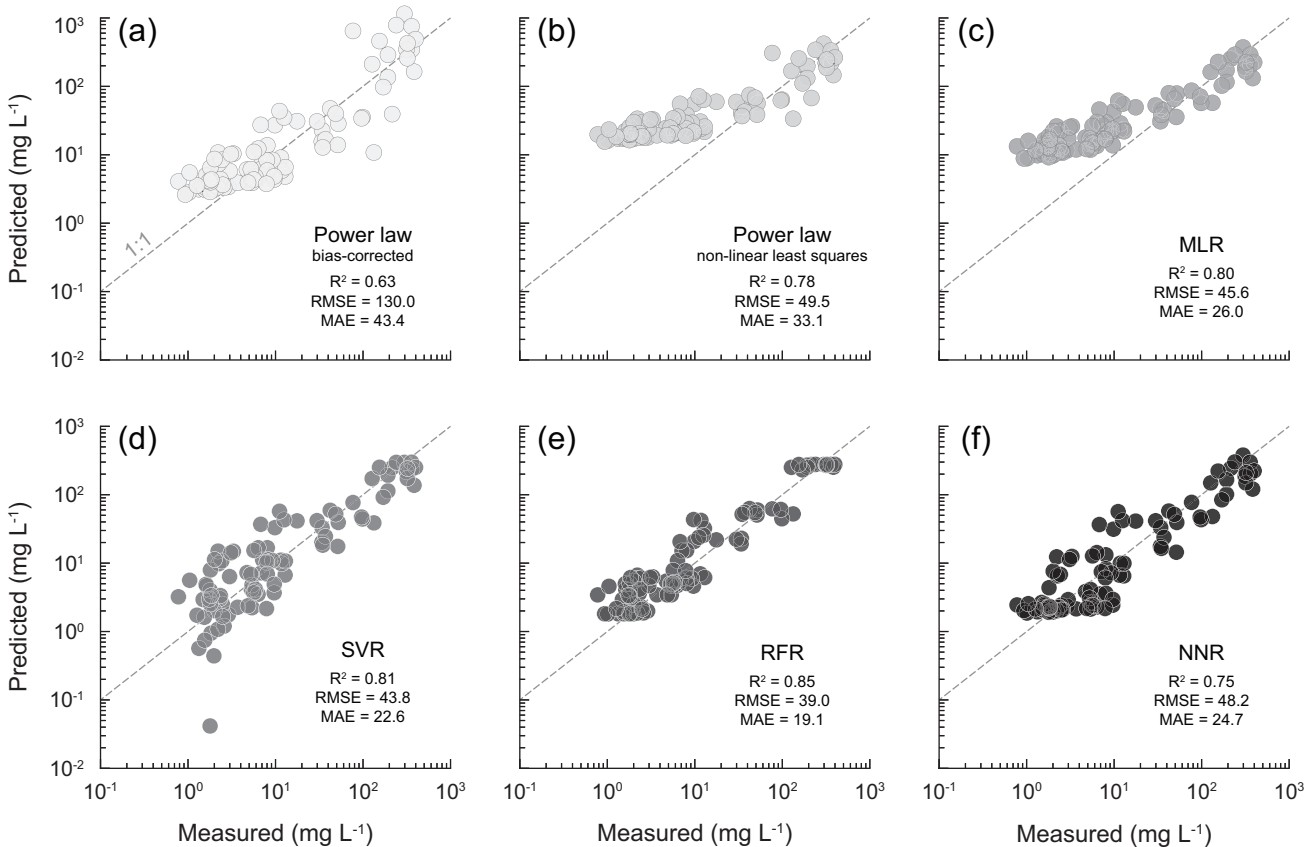


**Figure B1:** Performance of (a-b) traditional power law, (c) multiple linear regression (MLR), (d) support vector regression (SVR), (e) random forest regression (RFR), (f) neural network regression (NNR) models. The evaluation is based on observed against predicted suspended sediment values (mg L$^{-1}$). Performance metrics are based on nested cross-validation (R$^2$: coefficient of determination, RMSE: root mean squared error, MAE: mean absolute error).





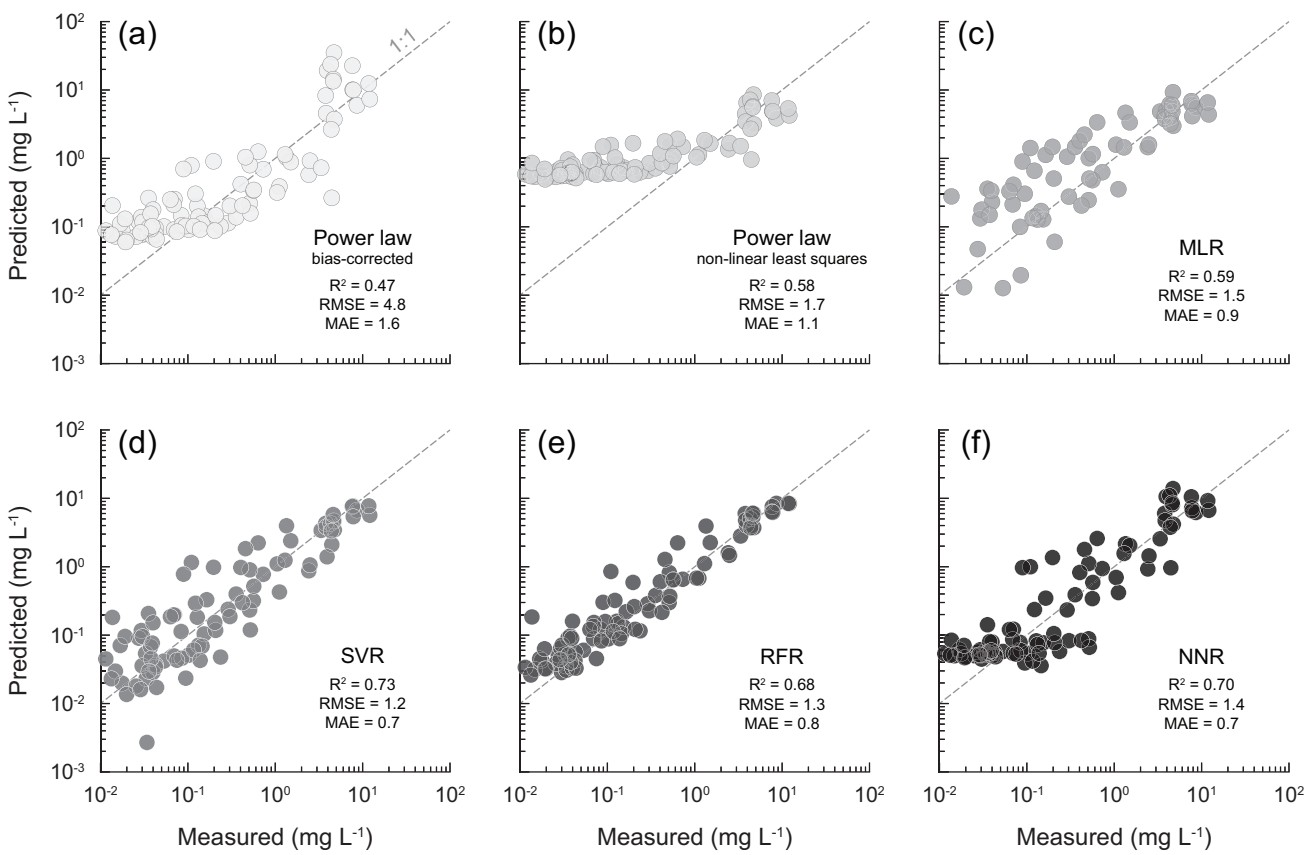

**Figure B2:** Performance of (a-b) traditional power law, (c) multiple linear regression (MLR), (d) support vector regression (SVR), and (e) random forest regression (RFR), (f) neural network regression (NNR) models. The evaluation is based on observed against predicted particulate organic carbon values (mg L⁻¹). Performance metrics are based on nested cross-validation ($R^2$: coefficient of determination, RMSE: root mean squared error, MAE: mean absolute error).

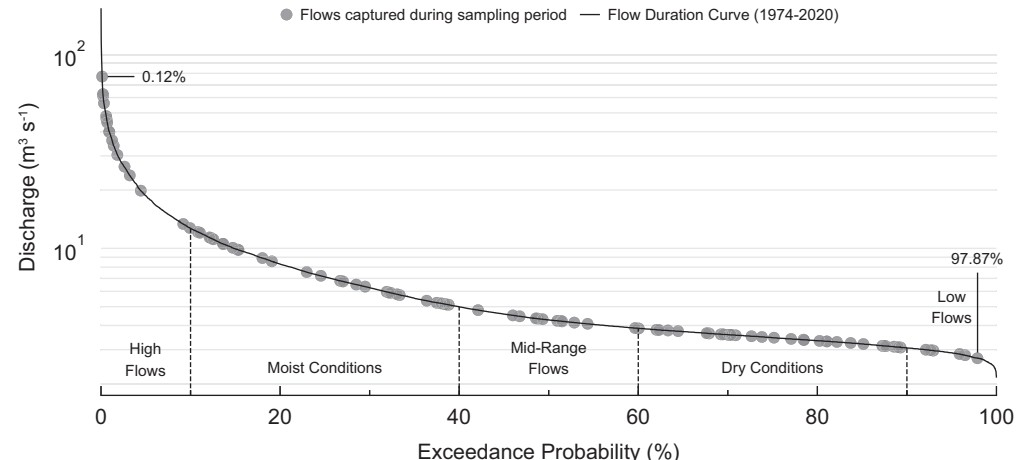

**Figure C1:** Flow duration curve of the Sihl River spanning from 1974 to 2020. Grey data points indicate sampling points.