# Peer review of "Environmental and hydrologic controls on sediment and organic carbon export from a subalpine catchment: insights from a time-series"

_EGUsphere, 2022_

## Referee Comment (RC2)

Review of "Environmental and hydrologic controls on sediment and organic carbon export from a subalpine catchment: insights from a time-series" by Melissa Schwab and co-authors.

**Summary:**

This manuscript presents a large dataset from a long-term sampling campaign of river water and suspended sediment from a subalpine catchment in Switzerland. The authors generated a substantial amount of data, including a 40-month time-series of stable carbon isotopes and radiocarbon activity of dissolved and particulate organic carbon. Time-series data sets like this are incredibly valuable to the scientific community, particularly now as our field aims to mechanistically describe the feedbacks between climate change and the global carbon cycle. This manuscript addresses relevant scientific questions (i.e., what controls the magnitude and temporal variability of river organic carbon export?). The main dataset and introduction of statistical approaches are a great contribution to the field. The methods and statistical analyses used in this manuscript are state of the art, particularly the application of EA-IRMS for high throughput 14C measurements and the application of machine learning-based statistical analyses.

Overall, the manuscript is well-written, but there several points that need clarification and revision, as noted in the major points of concern and the detailed comments below. The authors do a nice job of presenting their data and using statistics to describe the distribution of the data, however, it seems that this manuscript is lacking robust interpretation of the statistical results. Based on the introduction of the paper, I expected their results to provide a mechanistic explanation linking geomorphic and hydrologic processes to organic carbon export from small headwater rivers. However, I was not able to take away any new ideas or significant conclusions from their data interpretation and discussion. Additionally, I find that some of the analyses are not entirely appropriate (i.e. the MixSIAR analysis) and should be either removed from the manuscript or redone to reflect appropriate endmember mixing. To make this manuscript of greater interest to the scientific community, the authors should also provide a framework for integrating their statistical results into Earth system models.

In summary, there are a number of issues that need to be addresses before this manuscript can be accepted for publication in *EGU Biogeosciences*. Major points of concern and suggestions for revising the manuscript are detailed below.

**Major points of concern:**

The MixSiAR analysis is not applied appropriately here. Based on Figure 5a, two of the endmembers (soil and vegetation) are statistically indistinguishable such that they cannot be separate endmembers organic carbon sources. The authors constrain the soil endmember using samples from 0-40 cm soil depth, however, 0-10 cm soil typically contains a young organic horizon, such that its isotope composition will reflect modern vegetation. In lines 325-335, the authors explain that 14C-enriched bomb-derived OC can percolate through the soil column with DOC, causing homogenous isotopic composition with soil depth, however, the water-soluble phase of soil carbon should constitute only a minor portion of soil OC, such that it is unlikely for

this young OC pool to dominate the 14C activity of the soil profile. If the authors want to use soil as a distinct endmember in MixSIAR, I suggest they exclude soil data from <10 cm soil depth. Otherwise, they should perform a two end-member mixing model with biospheric and petrogenic OC as their end-member sources. Given that lake water/sediment samples are included in this study, perhaps an aquatic primary production endmember should be considered.

The authors do not discuss any underlying assumptions about the river system and its natural versus engineered state. I suspect that the response of suspended sediment and POC fluxes in the Sihl River will be dampened or altered by river engineering. The authors need to address the underlying assumptions of their study with respect to natural versus engineered rivers. They write in Line 88-90 that there are indeed weirs in the Sihl catchment that can trap 62-67% of the sediment, yet this effect is not addressed in the discussion.

The flow duration curve is an important aspect of this study, as it is used to define baseflow versus storm flow, however, I think this could be highlighted better in the main text and in a main figure. Figure C1 shows that the sample set covers nearly the entire range of discharge values, which is fantastic and shows how robust the statistical analyses can be. How does exceedance probability correlate to POC and DOC export and isotopic compositions? Instead of plotting run-off on the x-axis of Figure 7, would it be interesting to plot exceedance probability for the discharge recorded for each sample?

There are many details in the discussion section that are used to explain the data, but do not successfully build on the statistical results to make robust interpretations for how this analysis can be used to fill gaps in Earth system models. There are a lot of citations of previously published studies, although I feel that this is where new ideas should eb presented. This dataset and the statistical results can be used to predict carbon fluxes in Earth system models, something which we currently lack. I feel that the discussion and conclusions should be less focused on defending the data and more focused on developing a framework for including the predictive relationships in Earth system models.

**Detailed comments:**

L19: What aspect of petrogenic OC decreases downstream? Concentration or relative abundance?

L20: Changes in the relative proportions of OC sources and "overprinting" of isotopic signatures does NOT illustrate rapid OC transformation. The term "transformation" implies a change in chemical composition of individual organic molecules or particles (e.g., oxidation to CO2). Here, the authors are describing a dilution of one organic carbon source with respect to the total organic carbon, which is not a transformation.

L22-23: This sentence should be re-organized. As it is written, it reads as though the authors are saying that storms trigger surface runoff and shallow landslides, which is an obvious process

and not what their data are testing. Rather, the sentence could be written as "Our data suggest that storms enhance mobilization of fresh leaf litter and shallow soils via increased surface runoff and shallow landslides.'

L24: "Diverging mobilization pathways" is quite unclear here. The following sentence suggests that suspended sediment and POC mobilization are both related to water stage. And 1-day antecedent precipitation and discharge are both related to runoff, so it is unclear how these reflect different mobilization pathways. Perhaps the authors could interpret their statistical results to provide a mechanistic explanation of how 1-day antecedent moisture conditions affect POC in a different way than suspended sediment, as well as why discharge and suspended sediment concentration are more strongly linked than discharge and POC content.

Given the amount of data included in this study, I would expect more robust interpretations of the data than what is highlighted here in the abstract. Are there any events captured in the dataset that can be analyzed to explain the effects of high runoff or landslides on POC export?

L113: It is better to report discharge in common units of m3/s

L115: I suggest writing "...while bedrock lithologies in the ... are predominantly calcareous sandstones" rather than "bedrocks"

L122: Why was this sampling location chosen? Based on the coordinates provided, this stretch of river is heavily engineered, which may be both good and bad for this study. The channelization of the river may lead to efficient export of sediment from the upper catchment, however, this site is downstream of several dam-like structures that can retain sediment. How are the collected samples influenced by river engineering? Is there a portion of POC that could be missing due to sediment retention behind dams? Is river discharge also regulated by these dams and weirs, such that storm pulses are dampened and the effects of storms on POC export not fully captured? These issues should be addressed either here in the sample collection section or in the discussion section.

L129-130: What is meant by identical replicates? Did the authors fill three separate sampling containers for each sample? Was each water sample filtered with three separate glass fiber filters? If there were three separate filters, how was it ensured that the sampled water was homogeneous with respect to sediment concentration? Please clarify what is meant by replicates.

L133: were the DOC samples stored at 4 degrees C? Clarify the term "cooled."

L140-143: Were there no process blanks run to correct for possible carbon contamination introduced through filtration and sample handling? It is good practice to run process blanks in geochemistry.

L147: How was the CO2 captured and stored? Were CO2 sample vessels evacuated sufficiently prior to collecting evolved CO2? Was there a process blank for the sodium persulfate addition and helium purging steps?

L145: Were DOC concentrations measured? If yes, add this procedure to the methods section. If not, how did the authors know how much carbon was in their aliquots?

L156: Start this paragraph by saying what MixSIAR was used for and why before defining what the model is and how it was parameterized.

L165: What outputs were used from the MixSIAR model and how? Did the authors use the full posterior distbrutions in the analysis, the median, the mean and standard deviations of the posterior distributions? Please clarify and explain how the model output was interpreted.

L169: What is meant by "discern" here? This sentence is unclear. Do the authors mean that particulate matter concentrations tend to correlate with water discharge? If so, please be more clear.

L169: Where does the discharge data come from?

L170: by "insufficient" do the authors mean too infrequent? Be more specific.

L175-177: Is there an equation for the bias correction factor? Is the correction factor a nonlinear least squares regression approach? The last sentence is not well-integrated with the previous sentence. How does the correction affect the results? Is underestimation reduced or completely eliminated?

L179: Introduce why machine learning techniques are applied here. It is unclear why fluvial loads need to be estimated when there is likely a gauging station with hydrologic data that can be paired with samples. Please provide an explanation of what the authors expect to achieve with the machine learning analyses and why different types of analyses need to be compared.

L187-189: Why were these predictor variables chosen? Are there other studies that have shown these to be statistically useful parameters for predicting river particulate matter fluxes? If yes, please cite these references here.

L320-324: Based on the data plotted in Figure 5, it appears that there is either an older soil OC source or aquatic biomass source. The authors need to propose a mechanism for producing the isotope values that do not plot between their vegetation, soil, and bedrock endmembers. If they are going to rule out autochthonous production, then they need to provide an alternate hypothesis.

L325-335: The authors should provide stronger evidence for combining all soil data from 0-40 cm, as water-soluble organic carbon is only a fraction of total soil organic carbon and likely does not have a large effect on soil carbon isotope values. I would expect larger isotopic differences

between mineral soil and the organic horizon, which should be mobilized under different hydrologic/erosion conditions.

L500-501: None of the figures in this manuscript demonstrate that the authors have found new insights into temporal variations in OC export from their study catchment. Discharge and precipitation time-series are plotted in Figure 2, but the other figures primarily show the distribution of data across seasons, or isotope mixing diagrams that show no statistically significant correlations. I question whether the data support the authors' conclusions.

L495-498: In Figure 2, there is surprisingly little variability in the POC concentration and 14C activity of the time-series samples. The authors should elaborate more on the potential role of dams, weirs, and other channel engineering methods in regulating POC export from rivers.

**Figures and Tables:**

**Figure 2:**

- It is unconventional to order the plots from bottom to top. I suggest re-labeling them with plot (a) at the top and (h) at the bottom. It was confusing to read the caption and see hydrograph listed first.
- Place the letter labels in the same position on each plot (e.g., upper left corner).
- The circles with sizes representing the discharge during sampling should be at the top of the plot, rather than the bottom. As it is now, the small legend seems to correspond only to the discharge plot, making it difficult to figure out what the circles represent.

**Figure 5a:** The soil and vegetation data are statistically indistinguishable, making the MixSIAR three end-member mixing model invalid for this dataset. It is clear that there are other endmember POC sources that have lower d13C values and lower F14C values than the swiss soil data plotted here. Based on the lake values, aquatic productivity may be an important organic carbon endmember in the Sihl.

**Figure 5b:** This figure only shows that DOC is younger than POC for nearly all samples, which is common across many river systems. It's not clear why this plot is significant because DOC and POC have different sources.

**Table 2:**

- For the MixSIAR model input, there needs to be a number of samples (n) for each isotope tracer. In the table, n for F14C lists "est." The authors need to list what number was actually used, or put a note in the table footer.
- Need to insert table footer defining the variables (POC-d13C, POC-F14C, n, M, and SD).

**Figure 7:** There is not a clear purpose for including this figure in the main text. What are the gray arrows indicating on plots b and c? There are no statistics reported, yet it seems like the authors are proposing some sort of linear trend here.

---

## Author Comment (AC1)

**RESPONSE TO COMMENTS**

**COMMENTS FROM REVIEWER #2**

This manuscript presents a large dataset from a long-term sampling campaign of river water and suspended sediment from a subalpine catchment in Switzerland. The authors generated a substantial amount of data, including a 40-month time-series of stable carbon isotopes and radiocarbon activity of dissolved and particulate organic carbon. Time-series data sets like this are incredibly valuable to the scientific community, particularly now as our field aims to mechanistically describe the feedbacks between climate change and the global carbon cycle. This manuscript addresses relevant scientific questions (i.e., what controls the magnitude and temporal variability of river organic carbon export?). The main dataset and introduction of statistical approaches are a great contribution to the field. The methods and statistical analyses used in this manuscript are state of the art, particularly the application of EA-IRMS for high throughput 14C measurements and the application of machine learning-based statistical analyses.

Overall, the manuscript is well-written, but there several points that need clarification and revision, as noted in the major points of concern and the detailed comments below. The authors do a nice job of presenting their data and using statistics to describe the distribution of the data, however, it seems that this manuscript is lacking robust interpretation of the statistical results. Based on the introduction of the paper, I expected their results to provide a mechanistic explanation linking geomorphic and hydrologic processes to organic carbon export from small headwater rivers. However, I was not able to take away any new ideas or significant conclusions from their data interpretation and discussion. Additionally, I find that some of the analyses are not entirely appropriate (i.e. the MixSIAR analysis) and should be either removed from the manuscript or redone to reflect appropriate endmember mixing. To make this manuscript of greater interest to the scientific community, the authors should also provide a framework for integrating their statistical results into Earth system models.

In summary, there are a number of issues that need to be addresses before this manuscript can be accepted for publication in EGU Biogeosciences. Major points of concern and suggestions for revising the manuscript are detailed below.

**Response:** We thank the reviewer for the insightful comments and the valuable improvements to our manuscript. We address the raised concerns in detail below.

**Major points of concern:**

**Comment 1:** The MixSiAR analysis is not applied appropriately here. Based on Figure 5a, two of the endmembers (soil and vegetation) are statistically indistinguishable such that they cannot be separate endmembers organic carbon sources. The authors constrain the soil endmember using samples from 0-40 cm soil depth, however, 0-10 cm soil typically contains a young organic horizon, such that its isotope composition will reflect modern vegetation. In lines 325-335, the authors explain that 14C-enriched bomb-derived OC can percolate through the soil column with DOC, causing homogenous isotopic composition with soil depth,

however, the water-soluble phase of soil carbon should constitute only a minor portion of soil OC, such that it is unlikely for this young OC pool to dominate the 14C activity of the soil profile. If the authors want to use soil as a distinct endmember in MixSIAR, I suggest they exclude soil data from <10 cm soil depth. Otherwise, they should perform a two end-member mixing model with biospheric and petrogenic OC as their end-member sources. Given that lake water/sediment samples are included in this study, perhaps an aquatic primary production endmember should be considered.

**Response:** van der Voort et al. (2019) observed bulk soil organic matter with near atmospheric radiocarbon isotopic composition at depths larger than 40 cm in different Swiss study sites. They proposed the infiltration and assimilation of dissolved organic carbon containing bomb-derived radiocarbon as a likely mechanism to couple surface and deeper soil organic carbon pools. In addition, modern carbon might also be introduced by roots and mycorrhizal communities. However, we agree with the reviewer and divide the soil endmember based on the 10 cm threshold instead of 5 cm increments.

As suggested by the reviewer we have made modifications to our endmember compositions. (1) We utilize atmospheric radiocarbon averaged over the sampling period (May 2014 to March 2019; Hua et al., 2022) as source composition for vegetation endmember (wood, foliage). (2) Radiocarbon values of soils collected in 1998 are adjusted to the sampling period using turnover calculations and 2-point radiocarbon measurements (van der Voort et al., 2019). (3) Individual stable and radiocarbon endmember compositions are weighted by their organic carbon contents. (4) As sources are not significantly different either in stable or radiocarbon isotopic compositions, foliage, wood, top soils (<10 cm), and deep soils (> 10 cm) are combined into a biospheric endmember a priori before the mixing analysis. Despite these modifications to the endmember compositions, estimated proportions for biospheric and petrogenic contributions are unchanged. We extensively revised the manuscript to accommodate the described alterations.

We thank the reviewer for suggesting to include an aquatic primary production endmember. However, we consider aquatic in-situ production as an insignificant organic carbon source due to low levels of dissolved nutrients (Känel et al., 2021) and extensive shading in forested areas. Particulate organic carbon contributions from Lake Sihl are minimal as the reservoir retains 93% of the sediment (Grill et al., 2019). While the inflow of the Sihl River (Lake Sihl 1) displays similar particulate organic carbon isotopic compositions than the lower Sihl River, Lake Sihl stations 2 and 3 are more depleted in stable and radiocarbon values suggesting microbial activity and the metabolization of entrained organic carbon within the water column. If aquatic productivity constitutes a major endmember, we would expect similar organic carbon isotopic compositions in the lower Sihl River concurrent with phytoplankton blooms during the summer months. However, we observe an opposing trend as particulate organic carbon collected in summer is statistically more enriched in $^{13}C$ than compared to other seasons. Freshwater phytoplankton is highly variable in its carbon isotope composition (Chikaraishi, 2013). Without proper isolation and characterization, we cannot assign a profound endmember composition for primary production.

**Comment 2:** The authors do not discuss any underlying assumptions about the river system and its natural versus engineered state. I suspect that the response of suspended sediment and POC fluxes in the Sihl River will be dampened or altered by river engineering. The authors need to address the underlying assumptions of their study with respect to natural versus engineered rivers. They write in Line 88-90 that there are indeed weirs in the Sihl catchment that can trap 62-67% of the sediment, yet this effect is not addressed in the discussion.

**Response:** We agree with the reviewer to further elaborate on the role of artificial barriers along the Sihl River and have implemented a description in section 2.1 and extended our discussion of potential implications of man-made structures on the discharge of water and sediment in section 4.4.

Lake Sihl is an effective sediment trap retaining up to 93% of the sediment entrained from the upper catchment (Grill et al., 2019). The impact of the reservoir decreases from 93 % to 67 % and 62 % in the lower reaches of the Sihl River (Grill et al., 2019). The river course in the lower watershed, between Lake Sihl and the sampling location in Allmend Park, is fragmented by 4 low-head run-off-the-river hydroelectric systems and 14 weir structures.

Run-off-the-river systems divert and route portions of river water through a penstock towards hydroelectric turbines. The water is then returned to the river. In comparison to dams, run-off-the-river hydroelectric systems use the natural flow rate of the river and have little to no water storage capacity, posing minor obstructions to water and sediment export. Studies acknowledged that run-off-the-river can significantly impact sediment transport (Anderson et al., 2015). However, satellite imagery (Maxar Technologies) shows that low-head impoundments along the Sihl River result in a minimal rise of the hydraulic head and the storage of rather small water volumes. Headponds are likely to quickly fill with sediment ensuring the near-natural transport of sediment over the weir crest.

Weir structures consist mostly of broad-crested weirs, boulder weirs, and water stairs. Weirs or overflow dams are barriers that do not exceed the elevation of the top of the channel banks allowing constant flow over the weir crest during baseflow conditions (Csiki and Rhoads, 2010). Opposing to dams, weirs control the water level and river flow characteristics and do not act as reservoirs. A recent paper by Peeters et al. (2020), furthermore, argues that weirs are often leaky and inefficient in retaining bedload transport.

The Sihl River is characterized by a cobble and boulder river bed and low water levels (< 1 m). Storm-driven events lead to a quick rise in the water level, while limited floodplain extent (<10 %, Grill et al., 2019) and partly fortified river banks will support the pipping of water masses downstream. Although low-head run-off-the-river and weir structures might reduce the flow speed by diverting water masses, they do not store critical amounts of water and sediment during high discharge events and thus present no effective barriers against flooding.

**Comment 3:** The flow duration curve is an important aspect of this study, as it is used to define baseflow versus storm flow, however, I think this could be highlighted better in the main text and in a main figure. Figure C1 shows that the sample set covers nearly the entire range of discharge values, which is fantastic and shows how robust the statistical analyses can be. How does exceedance probability correlate to POC and DOC export and isotopic compositions? Instead of plotting run-off on the x-axis of Figure 7, would it be interesting to plot exceedance probability for the discharge recorded for each sample?

**Response:** We thank the reviewer for this suggestion but we respectfully disagree. Plotting particulate organic carbon contents, stable and radiocarbon isotopic compositions against exceedance probability will result in emphasizing baseflow conditions and marginalizing storm-driven events (Figure A). By definition, discharges ranging from low flow to moist conditions are captured as 90 % of the exceedance probability and represent flows from 2.17 to 12.7 $m^3$ $s^{-1}$. In contrast, high flow conditions consist of discharges from 12.7 to 77 (sampled discharges) to 172 $m^3$ $s^{-1}$ (discharge record 1974-2020). Our results indicate that baseflow conditions (<12.7 $m^3$ $s^{-1}$) account for only 18 % of water, 7.0 to 27.1 % of sediment, and 6.0 to 34.0 % of particulate organic carbon export. We favor the plotting of runoff (mm $d^{-1}$) which provides a more balanced representation of the data set. Plotting

runoff on the x-axis allows the visualization of potential changes in carbon content and isotopic composition as a function of discharge. It enables the direct comparison to Sihl River headwater streams as discharge records for the Erlenbach, Lümpenbach, and Vogelbach are not extensive enough to ensure conclusive flow duration curves.

[Figure]

**Figure A:** Relationship between exceedance probability (%), (a) particulate organic carbon (POC) content (wt%), (b) POC-$\delta^{13}$C (‰), and (c) POC-$F^{14}$C. Circles are color-coded for seasons.

**Comment 4:** There are many details in the discussion section that are used to explain the data, but do not successfully build on the statistical results to make robust interpretations for how this analysis can be used to fill gaps in Earth system models. There are a lot of citations of previously published studies, although I feel that this is where new ideas should eb presented. This dataset and the statistical results can be used to predict carbon fluxes in Earth system models, something which we currently lack. I feel that the discussion and conclusions should be less focused on defending the data and more focused on developing a framework for including the predictive relationships in Earth system models.

**Response:** While we appreciate the reviewer's feedback, we respectfully disagree. Our study combines a 40-month time series with novel approaches utilizing machine learning, environmental, and satellite records. We think this study makes a valuable contribution to the field as we observe a rapid change in the relative abundance of petrogenic and biospheric organic carbon within a short river segment. Large variations in the organic carbon isotopic compositions during baseflow and stormflow conditions suggest the mobilization and entrainment of different organic carbon pools. In addition to traditional rating curves, we apply machine learning approaches to estimate sediment and particulate organic carbon fluxes. To our knowledge, we are the first to assess fluvial organic carbon export using machine learning. Our results suggest that water stage and 1-day antecedent precipitations are important variables in predicting suspended sediment and particulate organic carbon fluxes.

We agree with the reviewer that applying our time series data set to infer improvements for Earth system models would be helpful. We show that discharge is insufficient as a sole parameter to characterize sediment and particulate organic carbon fluxes. Water stage, precipitation, and 1-day antecedent precipitation (and likely soil moisture) should be incorporated into model approaches. However, the scope of our manuscript is processes-based and aims to estimate sediment and particulate organic carbon loads for the Sihl River using statistical techniques, discuss the geochemical variability of organic carbon in response to changes in seasons and hydrology, and assess the role of the river as a link between headwater streams and low land rivers.

**Detailed comments:**

**Comment 5:** L19: What aspect of petrogenic OC decreases downstream? Concentration or relative abundance?

**Response:** Thank you for pointing this out. The relative abundance of petrogenic organic carbon decreases downstream. We have added this distinction to the abstract.

**Comment 6:** L20: Changes in the relative proportions of OC sources and "overprinting" of isotopic signatures does NOT illustrate rapid OC transformation. The term "transformation" implies a change in chemical composition of individual organic molecules or particles (e.g., oxidation to $CO_2$). Here, the authors are describing a dilution of one organic carbon source with respect to the total organic carbon, which is not a transformation.

**Response:** We agree with the reviewer's assessment and have exchanged "transformation" with "the rapid dilution of petrogenic OC over short distances".

**Comment 7:** L22-23: This sentence should be re-organized. As it is written, it reads as though the authors are saying that storms trigger surface runoff and shallow landslides, which is an obvious process and not what their data are testing. Rather, the

sentence could be written as "Our data suggest that storms enhance mobilization of fresh leaf litter and shallow soils via increased surface runoff and shallow landslides.

**Response:** As suggested by the reviewer, we have reformulated the sentence.

**Comment 8:** L24: "Diverging mobilization pathways" is quite unclear here. The following sentence suggests that suspended sediment and POC mobilization are both related to water stage. And 1-day antecedent precipitation and discharge are both related to runoff, so it is unclear how these reflect different mobilization pathways. Perhaps the authors could interpret their statistical results to provide a mechanistic explanation of how 1-day antecedent moisture conditions affect POC in a different way than suspended sediment, as well as why discharge and suspended sediment concentration are more strongly linked than discharge and POC content. Given the amount of data included in this study, I would expect more robust interpretations of the data than what is highlighted here in the abstract. Are there any events captured in the dataset that can be analyzed to explain the effects of high runoff or landslides on POC export?

**Response:** This time series was not designed to investigate individual storm-driven floodings or to tie particulate organic carbon export to specific geomorphological occurrences such as landslides. The field laboratory was intended to capture potential patterns across storm events, compare those patterns with baseflow conditions, and broadly infer potential mobilization and transport mechanisms of sediment and particulate organic carbon. Compositional, isotopic, and statistical evidence suggest that sediment and particulate organic carbon differ in their preferential sources and pathways in relation to discharge conditions. Particulate organic carbon exported during storm-driven events is characterized by more homogeneous stable carbon isotopic compositions and is enriched in $^{14}C$ than in comparison to samples collected during baseflow conditions. This behavior suggests the tapping of an organic carbon source that is preferentially exported during storms such as coarse plant debris that requires a certain runoff strength to be mobilized. Daily precipitation (including 1-day and 2-day antecedent precipitation) is a measure of the intensity of rainfall and can be interpreted as a potential driver for particle transport to adjacent aquatic systems. Precipitation and discharge are related to runoff. However, both parameters are not interchangeable as they behave differently in response to the duration of rain events, spatial shifts in precipitation strength, catchment morphology, and soil water saturation/soil drainage. In the Sihl River watershed, we observe a low correlation between daily discharge and daily precipitation ($r = 0.30$, $p < 0.001$, $n = 17167$; 1-day antecedent precipitation: $r = 0.06$, $p < 0.001$, $n = 17167$). Similarly, discharge and water stage are not linearly correlated as the rise of the water level will decelerate as soon as the water level reaches river banks and the adjacent riparian zone. Water stage might therefore act as a proxy for inundation and the mobilization of organic carbon deposited in the floodplain. Based on these assumptions, the relationships between water stage, 1-day antecedent precipitation, and particulate organic carbon concentrations might suggest that flooding, precipitation-induced erosion, and potentially shallow landsliding are major mechanisms facilitating the export of coarse discrete organic carbon via detaching litter and surface soil. The dependence of sediment concentrations on water stage and discharge might indicate that the majority of carbon-poor sediment is likely sourced from riverbeds and banks.

We acknowledge that the provided discussion is inadequate to fully convey our interpretation regarding potential mobilization drivers and pathways and has been revised in the abstract and expanded in section 4.3.

**Comment 9:** L113: It is better to report discharge in common units of m3 /s

**Response:** Discharge values for alpine streams are often reported in L s$^{-1}$ (e.g., Smith et al., 2013; Turowski et al., 2016; Hilton et al., 2021). As these values are quoted in the manuscript, we utilized the unit given in the references. However, to further clarity for the reader, we have changed the units to "0.038 to 0.077 m$^3$ s$^{-1}$" in the manuscript.

**Comment 10:** L115: I suggest writing "...while bedrock lithologies in the ... are predominantly calcareous sandstones" rather than "bedrocks"

**Response:** Thank you for this suggestion. "Bedrocks" in line L115 are exchanged with "bedrock lithologies".

**Comment 11:** L122: Why was this sampling location chosen? Based on the coordinates provided, this stretch of river is heavily engineered, which may be both good and bad for this study. The channelization of the river may lead to efficient export of sediment from the upper catchment, however, this site is downstream of several dam-like structures that can retain sediment. How are the collected samples influenced by river engineering? Is there a portion of POC that could be missing due to sediment retention behind dams? Is river discharge also regulated by these dams and weirs, such that storm pulses are dampened and the effects of storms on POC export not fully captured? These issues should be addressed either here in the sample collection section or in the discussion section.

**Response:** The Sihl River time series was established as a field laboratory. The sampling location in Allmend Park was chosen based on its accessibility, safety, and positioning within the watershed. Allmend Park can easily be reached via public transportation from ETH within 20 minutes. The proximity of the sampling location further allowed a quick response time to episodic storm-driven events. Large segments of the Sihl river course are flanked by major traffic routes, with no sidewalks and little protection against oncoming traffic. The location in Allmend Park is situated near the confluence with the Limmat River (~2 km) and thus captures the majority of the Sihl River watershed, an intermediary between headwater streams and lowland rivers.

The flow of the lower Sihl River is regulated by weir structures and run-off-the-river hydroelectric systems. These artificial barriers do not rise above channel banks leading to only minor increases in the hydraulic head. The resulting impoundments have little to no water storage capacity, with little potential to mitigate flooding. We believe that these man-made structures have an insignificant impact on the export of water, sediment, and particulate organic carbon, while partly fortified banks and narrow floodplains likely aid the downstream transport of the suspended load. Refer for a more detailed discussion to comment 2. We now address river engineering in sections 2.1 and 4.4.

**Comment 12:** L129-130: What is meant by identical replicates? Did the authors fill three separate sampling containers for each sample? Was each water sample filtered with three separate glass fiber filters? If there were three separate filters, how was it ensured that the sampled water was homogeneous with respect to sediment concentration? Please clarify what is meant by replicates.

**Response:** Thank you for pointing out that the collected samples were wrongly addressed as replicates. We filled three separate glass fiber filters and averaged water and sediment concentrations to get a more robust representation of the suspended load. We have rephrased the sentence in lines 129-130.

**Comment 13:** L133: were the DOC samples stored at 4 degrees C? Clarify the term "cooled."

**Response:** The reviewer's assumption is correct. DOC samples were stored at 4ºC. We have revised the text to include the precise storage temperature.

**Comment 14:** L140-143: Were there no process blanks run to correct for possible carbon contamination introduced through filtration and sample handling? It is good practice to run process blanks in geochemistry.

**Response:** Several process blanks have been run and are incorporated in the evaluation of constant contamination.

**Comment 15:** L147: How was the CO2 captured and stored? Were CO2 sample vessels evacuated sufficiently before collecting evolved CO2? Was there a process blank for the sodium persulfate addition and helium purging steps? L145: Were DOC concentrations measured? If yes, add this procedure to the methods section. If not, how did the authors know how much carbon was in their aliquots?

**Response:** We used a wet chemical approach evolving and capturing $CO_2$ directly in precombusted 12 mL gas-tight exetainer vials. Samples were measured within hours to 1-2 days minimizing periods of storage. Sample vessels do not require evacuating as the ambient air and inorganic carbon are purged with helium prior to converting dissolved organic carbon to $CO_2$.

Due to the low concentration of dissolved organic carbon concentrations in the Sihl River, 20 mL of sample material was accumulated in an exetainer vial by repeated freeze-drying. The dissolved organic carbon was then reconstituted in Milli-Q water, oxidized using an acidified sodium persulfate solution (100 mL $H_2O$ + 4.0 g $Na_2S_2O_8$ + 200 µL of 85 % $H_3PO_4$), and purged with high-purity helium gas (Grade 5.0, 99.9999% pure, for 10 min) removing ambient air and inorganic $CO_2$. The sample was heated to 100ºC for 1 h converting dissolved organic carbon to $CO_2$ within the exetainer vial. Within 1 to 2 days, the exetainer vial was loaded into the carbonate handling system of the mini radiocarbon dating system (MICADAS, Ionplus) equipped with a gas-accepting ion source. The automated headspace sampling transfers the evolved $CO_2$ over a magnesium perchlorate trap onto a zeolite molecular sieve. Subsequently, the zeolite trap is heated and the released $CO_2$ is collected in a gas-tight syringe and diluted with helium. The He-$CO_2$ mixture is then continuously pressed onto a Ti target in the gas-accepting ion source. Please, refer for a detailed description to Lang et al. (2012, 2016) and Wacker et al. (2013). In addition to phthalic acid and sucrose standards, we measured process, Milli-Q water, and persulfate blanks which are considered in the correction for extraneous carbon.

Dissolved organic carbon concentrations were measured and are reported in Appendix A and Table S1. Samples masses reported in line 145 are obtained from the elementar analyzer (Vario Micro, Elementar) coupled to the MICADAS system.

**Comment 16:** L156: Start this paragraph by saying what MixSIAR was used for and why before defining what the model is and how it was parameterized.

**Response:** We now provide an introductory sentence presenting our aim to apply MixSIAR as a Bayesian modeling approach to separate the contributions of potential organic carbon sources of the particulate load.

**Comment 17:** L165: What outputs were used from the MixSIAR model and how? Did the authors use the full posterior distrbutions in the analysis, the median, the mean and standard deviations of the posterior distributions? Please clarify and explain how the model output was interpreted.

**Response:** As indicated in line 355 we reported the mean and standard deviation of the full posterior probability distributions and used the mean to calculate fluxes of endmember contribution of particulate organic carbon (lines 469-472).

**Comment 18:** L169: What is meant by "discern" here? This sentence is unclear. Do the authors mean that particulate matter concentrations tend to correlate with water discharge? If so, please be more clear.

**Response:** This sentence is more ambiguous than intended. River discharge is a key parameter determining the export of sediment and particulate organic carbon. We have adjusted the sentence accordingly.

**Comment 19:** L169: Where does the discharge data come from?

**Response:** Discharge data were retrieved from the gauging station Sihlhölzli, Zurich, operated by the Swiss Federal Office for the Environment (FOEN, https://www.hydrodaten.admin.ch) (lines 198-190). To avoid repetitions, we address predictor variables after listing all model approaches.

**Comment 20:** L170: by "insufficient" do the authors mean too infrequent? Be more specific.

**Response:** We agree with the reviewer and replace "insufficient" with "too infrequent".

**Comment 21:** L175-177: Is there an equation for the bias correction factor? Is the correction factor a non-linear least squares regression approach? The last sentence is not well-integrated with the previous sentence. How does the correction affect the results? Is underestimation reduced or completely eliminated?

**Response:** The equation for the Duan bias correction factor ($\varepsilon$) which removes the gross bias in estimates has been included in the manuscript:

$$\varepsilon = \frac{1}{N} \sum_{i=1}^{N} \exp\left(\ln\left(C(obs)\right) - \ln(aQ^b)\right)$$

with $C$ as the observed suspended sediment concentration (mg L$^{-1}$), $Q$ as the water discharge (m$^3$ s$^{-1}$), and the rating coefficients $a$ and $b$.

Commonly, discharge and sediment or particulate organic carbon concentrations are described by a power relationship. For the purpose of data fitting and the application of ordinary least squares regression, training data are transformed using natural logarithms to convert the model to its **linear** equivalent. Ordinary least squares function fits a linear function by minimizing the sum of the squares of the residuals. However, ordinary least squares in a log-log space introduces a systematic bias to the residuals. The retransformation into a power function thus often results in the overestimation of small values and the underestimation of large values. The Duan correction factor merely minimizes this bias. **Non-linear** least squares regression allows us to fit a power law function directly to the data without the need for logarithmic transformation and thus we don't encounter the bias of log-transformed residuals. But this approach also poses statistical problems as the assumption of homoscedasticity is often not met. We have carefully revised section 2.5.

**Comment 22:** L179: Introduce why machine learning techniques are applied here. It is unclear why fluvial loads need to be estimated when there is likely a gauging station with hydrologic data that can be paired with samples. Please provide an

explanation of what the authors expect to achieve with the machine learning analyses and why different types of analyses need to be compared.

**Response:** Machine learning techniques are applied to estimate annual sediment and particulate organic carbon fluxes for the Sihl River basin (line 186). Power rating curves are limited in predicting export fluxes as they rely largely on the relationship between discharge and the suspended load. Machine learning approaches allow the application of more sophisticated functions - multilinear, support vector, random forest, and neural network regressions – and the utilization of additional predictor variables such as water stage, precipitation, 1-day, and 2-day antecedent precipitation to predict sediment and particulate organic carbon export. Utilized supervised machine learning approaches outperform traditional approaches in predicting sediment and particulate organic carbon concentrations (compare section 3.4-5; Figure 4, B1-2; Table B1-2). As none of the applied functions succeed in comprehensively describing the natural variations observed in the Sihl River basin (Table B1), we report the minimum and maximum values for the modeled annual loads as well as best-fitting model results throughout the manuscript.

We used the same training dataset for linear, non-linear, and machine learning functions consisting of observed sediment (n = 94) and particulate organic carbon (n = 90) concentrations and their respective discharge values obtained from the Sihlhölzli gauging station operated by the Swiss Federal Office for the Environment (FOEN, https://www.hydrodaten.admin.ch). The machine learning training dataset is augmented with water stage, precipitation, 1-day, and 2-day antecedent precipitation values.

**Comment 23:** L187-189: Why were these predictor variables chosen? Are there other studies that have shown these to be statistically useful parameters for predicting river particulate matter fluxes? If yes, please cite these references here.

**Response:** Studies showed that utilized parameters are statistically useful for predicting river particulate matter fluxes and have been listed in lines 181 to 184. We have included these references in line 189.

Our results also indicate that usage of water stage, precipitation, and 1-day antecedent precipitation are critical predictor variables enhancing the performance of model approaches in predicting sediment and organic carbon concentrations (Figure 4, B1, and B2; Table B1). Soil moisture as a parameter for soil water saturation might also pose as a useful variable. However, satellite-based soil moisture data are short (2015-now), sparse (2-3-day returns), and patchy (missing data) to form a continuous record needed to estimate robust annual yields for sediment and particulate organic carbon loads.

**Comment 24:** L320-324: Based on the data plotted in Figure 5, it appears that there is either an older soil OC source or aquatic biomass source. The authors need to propose a mechanism for producing the isotope values that do not plot between their vegetation, soil, and bedrock endmembers. If they are going to rule out autochthonous production, then they need to provide an alternate hypothesis.

**Response:** Endmember compositions plotted in Figure 5a do not represent the full range of observed values but consist of the mean and one standard deviation (Table 2). Observed stable isotopic compositions for soils span values from -31.36 to -22.70 ‰ (Smith et al., 2013; van der Voort et al., 2016; Gies et al., 2022). Similarly, the vegetation endmember ranges from -31.37 to -22.70 ‰ (Smith et al., 2013; Gies et al., 2022). Based on these values, Sihl River samples fall within the proposed endmember compositions. The isotopic compositions of Lake Sihl samples are discussed in section 4.2. We have replaced the combined vegetation endmember with the original foliage and wood endmembers (Figure B).

Endmember compositions are limited by small sample sizes and are associated with large uncertainties how well these samples represent the vegetation and soils present in the watershed. The isotopic composition of the Erlenbach, Lümpenbach, and Vogelbach indicate vegetation, soil, and bedrock sources that are not captured in previous surveys. Another potential source might consist of carbon-poor bank and riverbed sediments. We assume aquatic primary productivity in headwater streams to be negligible due to limited light conditions/shading, steep gradients, turbulent flow, and short residence times.

[Figure]

Figure B: Relationship between (a) particulate organic carbon (POC) $F^{14}C$ and POC-$\delta^{13}C$ (‰) values. Gray crosses indicate samples from the Alptal headwater streams: Erlenbach, Lümpenbach, and Vogelbach (Smith et al., 2013; Gies et al., 2022). Mean isotopic compositions (±SD) of bedrock, foliage, wood, top and deep soil endmembers are depicted. Dots are color-coded for seasons and scaled to discharge.

**Comment 25:** L325-335: The authors should provide stronger evidence for combining all soil data from 0-40 cm, as water-soluble organic carbon is only a fraction of total soil organic carbon and likely does not have a large effect on soil carbon isotope values. I would expect larger isotopic differences between mineral soil and the organic horizon, which should be mobilized under different hydrologic/erosion conditions.

**Response:** Bulk organic matter in Swiss top and deep (up to 40 cm) soils displays an enrichment in bomb-derived radiocarbon (van der Voort et al., 2016, 2019). Water-extractable organic carbon has been introduced as a likely agent connecting shallow and deeper soil horizons. Please, refer to comment 1 for a more detailed discussion.

**Comment 26:** L500-501: None of the figures in this manuscript demonstrate that the authors have found new insights into temporal variations in OC export from their study catchment. Discharge and precipitation time-series are plotted in Figure 2, but the other figures primarily show the distribution of data across seasons, or isotope mixing diagrams that show no statistically significant correlations. I question whether the data support the authors' conclusions.

**Response:** While we appreciate the reviewer's feedback, we respectfully disagree. Our study combines a 40-month time series with novel approaches utilizing machine learning approaches, environmental and satellite records. We think this study makes a valuable contribution to the field as we observe a rapid change in the relative abundance of petrogenic and biospheric organic carbon within a short river segment. Large variations in the organic carbon isotopic compositions during baseflow and stormflow conditions suggest the mobilization and entrainment of different organic carbon pools. In addition to traditional rating curves, we apply machine learning approaches to estimate sediment and particulate organic carbon fluxes. To our knowledge, we are the first to assess fluvial organic carbon export using machine learning. Our results suggest that water stage and 1-day antecedent precipitations are important variables in predicting suspended sediment and particulate organic carbon fluxes.

**Comment 27:** L495-498: In Figure 2, there is surprisingly little variability in the POC concentration and 14C activity of the time-series samples. The authors should elaborate more on the potential role of dams, weirs, and other channel engineering methods in regulating POC export from rivers.

**Response:** We have added the suggested content to the manuscript in sections 2.1 and 4.4. Please, compare responses to comments 2 and 11.

**Figures and Tables:**

**Comment 28:** Figure 2:

- It is unconventional to order the plots from bottom to top. I suggest re-labeling them with plot (a) at the top and (h) at the bottom. It was confusing to read the caption and see hydrograph listed first.

- Place the letter labels in the same position on each plot (e.g., upper left corner).

- The circles with sizes representing the discharge during sampling should be at the top of the plot, rather than the bottom. As it is now, the small legend seems to correspond only to the discharge plot, making it difficult to figure out what the circles represent.

**Response:** We concur with the reviewer and implemented the proposed changes in the structure of the subplots, the positioning of the labels and legends.

**Comment 29:** Figure 5a: The soil and vegetation data are statistically indistinguishable, making the MixSIAR three end-member mixing model invalid for this dataset. It is clear that there are other endmember POC sources that have lower d13C values and lower F14C values than the swiss soil data plotted here. Based on the lake values, aquatic productivity may be an important organic carbon endmember in the Sihl.

**Response:** Endmember compositions displayed in figure 5a consist of mean and one standard deviation and do not represent the whole range of observed values. We believe that primary productivity is a minor component of the suspended particulate organic carbon pool as the Sihl River is characterized by low nutrient levels, extensive shading, and short transit times. Please refer to our responses to comments 1, 24, and 25.

**Comment 30:** Figure 5b: This figure only shows that DOC is younger than POC for nearly all samples, which is common across many river systems. It's not clear why this plot is significant because DOC and POC have different sources.

**Response:** We respectfully disagree with the reviewer. Figure 5b illustrates the predominant modern radiocarbon composition of exported particulate and dissolved organic carbon pools despite the impact of settlements and the extensively engineered landscape.

**Comment 31:** Table 2:

- For the MixSIAR model input, there needs to be a number of samples (n) for each isotope tracer. In the table, n for F14C lists "est." The authors need to list what number was actually used, or put a note in the table footer.

- Need to insert table footer defining the variables (POC-d13C, POC-F14C, n, M, and SD).

**Response:** The radiocarbon composition of bedrock samples has not been reported. We assume that the 22 characterized bedrock samples are radiocarbon-free (0.00±0.01). We have added an explanation to the caption.

We thank the reviewer for pointing out this oversight. We now include definitions for variables in Table 2 and B1. However, common statistical abbreviations used in Tables 2, B1, and D1 do not require explanations (APA style).

**Comment 32:** Figure 7: There is not a clear purpose for including this figure in the main text. What are the gray arrows indicating on plots b and c? There are no statistics reported, yet it seems like the authors are proposing some sort of linear trend here.

**Response:** Figure 7 displays the behavior of organic carbon content, stable and radiocarbon isotopic compositions plotted against the continuous variable runoff and allows a direct comparison to Sihl River headwater streams. Gray arrows indicate trends (line 427) in the isotopic composition of organic carbon. Stable organic carbon values appear to converge with higher discharges. As the means are similar for baseflow and stormflow conditions, this observation cannot be resolved with an analysis of variance. The grey arrow is supposed to guide the eye of the reader in perceiving the reduction in isotopic variability of the Sihl River and headwaters. Particulate organic radiocarbon values are significantly different during low and high flows and also show a significant monotonic increase with discharge ($r_s = 0.46$, $p<0.001$). We have included the spearman correlation in subplot 7c and a remark regarding the trendlines in the caption.

**References**

Anderson D., Moggridge H., Warren P. and Shucksmith J. (2015) The impacts of "run-of-river" hydropower on the physical and ecological condition of rivers. *Water and Environment Journal* **29**, 268–276.

Chikaraishi Y. (2013) 13C/12C Signatures in Plants and Algae. In *Treatise on Geochemistry: Second Edition* Elsevier Ltd. pp. 95–123.

Csiki S. and Rhoads B. L. (2010) Hydraulic and geomorphological effects of run-of-river dams. *Prog Phys Geogr* **34**, 755–780.

Gies H., Lupker M., Wick S., Haghipour N., Buggle B. and Eglinton T. (2022) Discharge-Modulated Soil Organic Carbon Export From Temperate Mountainous Headwater Streams. *J Geophys Res Biogeosci* **127**.

Grill G., Lehner B., Thieme M., Geenen B., Tickner D., Antonelli F., Babu S., Borrelli P., Cheng L., Crochetiere H., Ehalt Macedo H., Filgueiras R., Goichot M., Higgins J., Hogan Z., Lip B., McClain M. E., Meng J., Mulligan M., Nilsson C., Olden J. D., Opperman J. J., Petry P., Reidy Liermann C., Sáenz L., Salinas-Rodríguez S., Schelle P., Schmitt R. J. P., Snider J., Tan F., Tockner K., Valdujo P. H., van Soesbergen A. and Zarfl C. (2019) Mapping the world's free-flowing rivers. *Nature* **569**, 215–221.

Hilton R. G., Turowski J. M., Winnick M., Dellinger M., Schleppi P., Williams K. H., Lawrence C. R., Maher K., West M. and Hayton A. (2021) Concentration-Discharge Relationships of Dissolved Rhenium in Alpine Catchments Reveal Its Use as a Tracer of Oxidative Weathering. *Water Resour Res* **57**.

Hua Q., Turnbull J. C., Santos G. M., Rakowski A. Z., Ancapichún S., de Pol-Holz R., Hammer S., Lehman S. J., Levin I., Miller J. B., Palmer J. G. and Turney C. S. M. (2022) Atmospheric radiocarbon for the period 1950-2019. *Radiocarbon* **64**, 723–745.

Känel B., Götz C., Niederhauser P., Sinniger J. and Steinmann P. (2021) *Zustand der Fliessgewässer von Limmat, Sihl und Zürichsee - Messkampagne 2020.*,

Lang S. Q., Bernasconi S. M. and Früh-Green G. L. (2012) Stable isotope analysis of organic carbon in small (ug C) samples and dissolved organic matter using a GasBench preparation device. *Rapid Communications in Mass Spectrometry* **26**, 9–16.

Lang S. Q., Mcintyre C. P., Bernasconi S. M., Früh-Green G. L., Voss B. M., Eglinton T. I. and Wacker L. (2016) Rapid [14]C Analysis of Dissolved Organic Carbon in Non-Saline Waters. *Radiocarbon* **58**, 1–11.

Peeters A., Houbrechts G., Hallot E., van Campenhout J., Gob F. and Petit F. (2020) Can coarse bedload pass through weirs? *Geomorphology* **359**.

Smith J. C., Galy A., Hovius N., Tye A. M., Turowski J. M. and Schleppi P. (2013) Runoff-driven export of particulate organic carbon from soil in temperate forested uplands. *Earth Planet Sci Lett* **365**, 198–208.

Turowski J. M., Hilton R. G. and Sparkes R. (2016) Decadal carbon discharge by a mountain stream is dominated by coarse organic matter. *Geology* **44**, 27–30.

van der Voort T. S., Hagedorn F., McIntyre C., Zell C., Walthert L., Schleppi P., Feng X. and Eglinton T. I. (2016) Variability in [14]C contents of soil organic matter at the plot and regional scale across climatic and geologic gradients. *Biogeosciences* **13**, 3427–3439.

van der Voort T. S. van der, Mannu U., Hagedorn F., McIntyre C. P., Walthert L., Schleppi P., Haghipour N. and Eglinton T. I. (2019) Dynamics of deep soil carbon - insights from [14]C time series across a climatic gradient. *Biogeosciences* **16**, 3233–3246.

Wacker L., Fahrni S. M., Hajdas I., Molnar M., Synal H. A., Szidat S. and Zhang Y. L. (2013) A versatile gas interface for routine radiocarbon analysis with a gas ion source. *Nucl Instrum Methods Phys Res B* **294**, 315–319.

---

## Author Response (AR2)

**egusphere-2022-705**

**RESPONSE TO COMMENTS**

**Dear Dr. Naeher,**

Thank you for giving us the opportunity to submit a revised draft of the manuscript "*Environmental and hydrologic controls on sediment and organic carbon export from a subalpine catchment: insights from a time-series*" and for considering it for publication in *Biogeosciences*. We would like to express our sincere gratitude to you and both reviewers. Their careful and thorough reading of this paper resulted in insightful comments and constructive suggestions which helped us to improve the quality of the manuscript.

The following provides a point-by-point response to the reviewers' comments and concerns. Line numbers in black refer to the original manuscript, while line numbers in blue point to the revised manuscript

**COMMENTS FROM REVIEWER #1**

In this manuscript, the authors present a substantially long time-series of isotopic and hydrographic measurements to understand carbon cycling patterns along the Sihl River and basin. The strength of this manuscript is its clear goal and approach. The study uses a mixture of "traditional" methods to measure carbon cycling (POC and DOC collection via filtration, and thorough subsequent geochemical analysis (2.3)) and new computational methods such as machine learning. Many of the terms / methods described in 2.5 and 2.6 are new to me as someone largely unfamiliar with machine learning algorithms; however, this does not negatively impact the article's "result traceability", and there is significant precedence provided for each decision via citation of previous literature.

One minor comment / question (Line 132). I'm unfamiliar with this method of storing DOC samples, and am surprised they were not frozen. I'm wondering if the effects of acidification versus freezing was considered in the method, or interpretation of results? See Walker et al., 2016 https://doi.org/10.1017/RDC.2016.48

Overall, I find this manuscript very strong and support straightforward publication.

**Response:** We appreciate the encouraging comments and wish to thank the reviewer for the effort and time spent reviewing the manuscript.

Current literature provides several accepted protocols for handling and storing dissolved organic carbon. Storage and preservation methods include a variety of containers (borosilicate vs HDPE), biocides (e.g., HgCl2, NaN3, HCl, HNO3), and storage temperatures (refrigerated vs frozen). The commonly recommended methods are frozen storage (Walker et al., 2017;

Heinz and Zak, 2018) and the storage of acidified samples in cold temperatures (4° C) (Cook et al., 2016; Nachimuthu et al., 2020). However, both options are characterized by advantages and disadvantages. Thieme et al. (2016) and Walker et al. (2017) demonstrate that freezing might preserve dissolved organic carbon concentrations and isotopic compositions while Spencer et al. (2007) and Thieme et al. (2016) argue that freeze/thaw cycles likely affect chemical and optical compositions of dissolved organic matter. Opinions further differ regarding the pre-treatment of dissolved organic carbon for freshwater and marine water samples. Regardless of the preservation method, organic carbon will be subjected to decomposition and alteration with increasing storage time.

Due to the low concentrations of dissolved organic carbon concentrations in the Sihl River, 20 mL of sample material were concentrated in precombusted gas-tight 12 mL exetainer vials by repeated freeze-drying of 5 mL aliquots. The vials were stored frozen until further analysis. We have failed to include this preparation step in the methods and have revised section 2.3 (lines 143-145).

**COMMENTS FROM REVIEWER #2**

This manuscript presents a large dataset from a long-term sampling campaign of river water and suspended sediment from a subalpine catchment in Switzerland. The authors generated a substantial amount of data, including a 40-month time-series of stable carbon isotopes and radiocarbon activity of dissolved and particulate organic carbon. Time-series data sets like this are incredibly valuable to the scientific community, particularly now as our field aims to mechanistically describe the feedbacks between climate change and the global carbon cycle. This manuscript addresses relevant scientific questions (i.e., what controls the magnitude and temporal variability of river organic carbon export?). The main dataset and introduction of statistical approaches are a great contribution to the field. The methods and statistical analyses used in this manuscript are state of the art, particularly the application of EA-IRMS for high throughput 14C measurements and the application of machine learning-based statistical analyses.

Overall, the manuscript is well-written, but there several points that need clarification and revision, as noted in the major points of concern and the detailed comments below. The authors do a nice job of presenting their data and using statistics to describe the distribution of the data, however, it seems that this manuscript is lacking robust interpretation of the statistical results. Based on the introduction of the paper, I expected their results to provide a mechanistic explanation linking geomorphic and hydrologic processes to organic carbon export from small headwater rivers. However, I was not able to take away any new ideas or significant conclusions from their data interpretation and discussion. Additionally, I find that some of the analyses are not entirely appropriate (i.e. the MixSIAR analysis) and should be either removed from the manuscript or redone to reflect appropriate endmember mixing. To make this manuscript of greater interest to the scientific community, the authors should also provide a framework for integrating their statistical results into Earth system models.

In summary, there are a number of issues that need to be addresses before this manuscript can be accepted for publication in EGU Biogeosciences. Major points of concern and suggestions for revising the manuscript are detailed below.

**Response:** We thank the reviewer for the insightful comments and the valuable improvements to our manuscript. We address the raised concerns in detail below.

**Major points of concern:**

**Comment 1:** The MixSiAR analysis is not applied appropriately here. Based on Figure 5a, two of the endmembers (soil and vegetation) are statistically indistinguishable such that they cannot be separate endmembers organic carbon sources. The authors constrain the soil endmember using samples from 0-40 cm soil depth, however, 0-10 cm soil typically contains a young organic horizon, such that its isotope composition will reflect modern vegetation. In lines 325-335, the authors explain that 14C-enriched bomb-derived OC can percolate through the soil column with DOC, causing homogenous isotopic composition with soil depth, however, the water-soluble phase of soil carbon should constitute only a minor portion of soil OC, such that it is unlikely for this young OC pool to dominate the 14C activity of the soil profile. If the authors want to use soil as a distinct endmember in MixSIAR, I suggest they exclude soil data from <10 cm soil depth. Otherwise, they should perform a two end-member mixing model with biospheric and petrogenic OC as their end-member sources. Given that lake water/sediment samples are included in this study, perhaps an aquatic primary production endmember should be considered.

**Response:** van der Voort et al. (2019) observed bulk soil organic matter with near atmospheric radiocarbon isotopic composition at depths greater than 30 cm in different Swiss study sites, including Alptal. They proposed the infiltration and assimilation of dissolved organic carbon containing bomb-derived radiocarbon as a likely mechanism to couple surface and deeper soil organic carbon pools. In addition, modern carbon might also be introduced by roots and mycorrhizal communities. However, we agree with the reviewer and divide the soil endmember based on the 10 cm threshold instead of 5 cm increments.

As suggested by the reviewer we have made modifications to our endmember compositions. (1) We utilize atmospheric radiocarbon averaged over the sampling period (May 2014 to March 2019; Hua et al., 2022) as source composition for vegetation endmember (wood, foliage). (2) Radiocarbon values of soils collected in 1998 are adjusted to the sampling period using turnover calculations and 2-point radiocarbon measurements (van der Voort et al., 2019). (3) Individual stable and radiocarbon endmember compositions are weighted by their organic carbon contents. (4) As sources are not significantly different either in stable or radiocarbon isotopic compositions, foliage, wood, top soils (<10 cm), and deep soils (> 10 cm) are combined into a single biospheric endmember a priori before the mixing analysis. Despite these modifications to the endmember compositions, proportions for biospheric and petrogenic contributions reproduced previously computed estimations. We have extensively revised the manuscript to accommodate the described modifications.

We thank the reviewer for suggesting to include an aquatic primary production endmember. However, we consider aquatic insitu production as an insignificant organic carbon source due to short water transit times, low levels of dissolved nutrients (Känel et al., 2021) and extensive shading due to forested areas as well as frequently turbid waters (particularly during high flow when most sediment and carbon discharge occurs). Particulate organic carbon (POC) contributions from Lake Sihl are minimal as the reservoir retains 93% of the sediment (Grill et al., 2019). While the inflow of the Sihl River (Lake Sihl 1) displays similar POC isotopic compositions than the lower Sihl River, Lake Sihl stations 2 and 3 exhibit lower stable organic carbon and radiocarbon isotopic values suggesting microbial activity and the metabolization of entrained organic carbon within the water column. If aquatic productivity constitutes a major endmember, we would expect similar organic carbon isotopic compositions in the lower Sihl River concurrent with phytoplankton blooms during the summer months. However, we observe an opposing trend as POC collected in summer is statistically more enriched in 13C than compared to other seasons. Freshwater phytoplankton is highly variable in its carbon isotope composition (Chikaraishi, 2013). Without proper isolation and characterization, we cannot confidently assign an endmember composition for primary production.

**Comment 2:** The authors do not discuss any underlying assumptions about the river system and its natural versus engineered state. I suspect that the response of suspended sediment and POC fluxes in the Sihl River will be dampened or altered by river engineering. The authors need to address the underlying assumptions of their study with respect to natural versus engineered rivers. They write in Line 88-90 that there are indeed weirs in the Sihl catchment that can trap 62-67% of the sediment, yet this effect is not addressed in the discussion.

**Response:** We agree with the reviewer to further elaborate on the role of artificial barriers along the Sihl River and have implemented a description in section 2.1 and extended our discussion of potential implications of man-made structures on the discharge of water and sediment in section 4.4.

Lake Sihl is an effective sediment trap retaining up to 93% of the sediment entrained from the upper catchment (Grill et al., 2019). The impact of the reservoir on sediment retention decreases from 93 % to 67 % and 62 % in the lower reaches of the Sihl River (Grill et al., 2019). The river course in the lower watershed, between Lake Sihl and the sampling location in Allmend Park, is fragmented by 4 low-head run-off-the-river hydroelectric systems and 14 weir structures.

Run-off-the-river systems divert and route portions of river water through a penstock towards hydroelectric turbines. The water is then returned to the river. In comparison to dams, run-off-the-river hydroelectric systems use the natural flow rate of the river and have little to no water storage capacity, thus posing minor obstructions to water and sediment export. Studies acknowledged that run-off-the-river can significantly impact sediment transport (Anderson et al., 2015). However, satellite imagery (Maxar Technologies) shows that low-head impoundments along the Sihl River result in a minimal rise of the hydraulic head and the storage of rather small water volumes. Headponds are likely to quickly fill with sediment ensuring the near-natural transport of sediment over the weir crest.

Weir structures consist mostly of broad-crested weirs, boulder weirs, and water stairs. Weirs or overflow dams are barriers that do not exceed the elevation of the top of the channel banks, allowing constant flow over the weir crest during baseflow conditions (Csiki and Rhoads, 2010). In contrast to dams, weirs control the water level and river flow characteristics and do not act as reservoirs. A recent paper by Peeters et al. (2020), furthermore, argues that weirs are often leaky and inefficient in retaining bedload transport.

The Sihl River is characterized by a cobble and boulder river bed and generally low water levels (< 1 m). Storm-driven events lead to a quick rise in the water level, while limited floodplain extent (<10 %, Grill et al., 2019) and partly fortified river banks will support the piping of water masses downstream. Overall, while low-head run-off-the-river and weir structures might reduce the flow speed by diverting water masses, they do not store critical amounts of water and sediment during high discharge events and thus present no effective barriers against flooding.

**Comment 3:** The flow duration curve is an important aspect of this study, as it is used to define baseflow versus storm flow, however, I think this could be highlighted better in the main text and in a main figure. Figure C1 shows that the sample set covers nearly the entire range of discharge values, which is fantastic and shows how robust the statistical analyses can be. How

does exceedance probability correlate to POC and DOC export and isotopic compositions? Instead of plotting run-off on the xaxis of Figure 7, would it be interesting to plot exceedance probability for the discharge recorded for each sample?

**Response:** We thank the reviewer for this suggestion, but we respectfully disagree. Plotting POC contents, stable and radiocarbon isotopic compositions against exceedance probability will result in emphasizing baseflow conditions and marginalizing storm-driven events (Figure A). By definition, discharges ranging from low flow to moist conditions are captured as 90 % of the exceedance probability and represent flows from 2.17 to 12.7 m3 s-1. In contrast, high flow conditions consist of discharges from 12.7 to 77 (sampled discharges) to 172 m3 s-1 (discharge record 1974-2020). Our results indicate that baseflow conditions (<12.7 m3 s-1) account for only 18 % of water, 7.0 to 27.1 % of sediment, and 6.0 to 34.0 % of POC export. We favor the plotting of runoff (mm d-1) which provides a more balanced representation of the data set. Plotting runoff on the x-axis allows the visualization of potential changes in carbon content and isotopic composition as a function of discharge. It enables the direct comparison to Sihl River headwater streams as discharge records for the Erlenbach, Lümpenbach, and Vogelbach streams are not extensive enough to ensure conclusive flow duration curves.

**Comment 4:** There are many details in the discussion section that are used to explain the data, but do not successfully build on the statistical results to make robust interpretations for how this analysis can be used to fill gaps in Earth system models. There are a lot of citations of previously published studies, although I feel that this is where new ideas should eb presented. This dataset and the statistical results can be used to predict carbon fluxes in Earth system models, something which we currently lack. I feel that the discussion and conclusions should be less focused on defending the data and more focused on developing a framework for including the predictive relationships in Earth system models.

**Response:** We agree with the reviewer and hope the data presented in this study and associated statistical analyses can ultimately help inform Earth system models (ESMs). However, we consider that studies such as this remain to scarce for robust parameterization and extrapolation to other systems. Our study combines a 40-month time series with novel approaches utilizing machine learning, environmental, and satellite records. We think this study makes a valuable contribution to the field as we observe a rapid change in the relative abundance of petrogenic and biospheric organic carbon within a short river segment as a function of flow conditions, serving to bridge prior investigations of headwater catchments and large river systems. We believe we are amongst the first to assess fluvial organic carbon export using machine learning methods. Our results suggest that discharge is insufficient as a sole parameter to characterize sediment and POC fluxes and ought to be augmented by water stage and 1-day antecedent precipitations. We consider the scope of our manuscript is processes-based and aims to constrain the factors influencing sediment and POC loads and the composition of organic carbon for the Sihl River using statistical techniques. We feel further investigations are warranted before such observations can be used in a predictive fashion and incorporated into ESMs.

**Figure A:** Relationship between exceedance probability (%), (a) particulate organic carbon (POC) content (wt%), (b) POC- $\delta^{13}$ C (‰), and (c) POC- $F^{14}$ C. Circles are color-coded for seasons.

**Detailed comments:**

**Comment 5: L19: What aspect of petrogenic OC decreases downstream? Concentration or relative abundance?**

**Response:** Thank you for pointing this out. The relative amount of petrogenic organic carbon decreases downstream. We have added this distinction to the abstract (lines 17-18).

**Comment 6:** L20: Changes in the relative proportions of OC sources and "overprinting" of isotopic signatures does NOT illustrate rapid OC transformation. The term "transformation" implies a change in chemical composition of individual organic molecules or particles (e.g., oxidation to CO2). Here, the authors are describing a dilution of one organic carbon source with respect to the total organic carbon, which is not a transformation.

**Response:** We agree with the reviewer's assessment and have exchanged "transformation" with "the rapid organic matter alteration over short distances" (lines 19-20).

**Comment 7:** L22-23: This sentence should be re-organized. As it is written, it reads as though the authors are saying that storms trigger surface runoff and shallow landslides, which is an obvious process and not what their data are testing. Rather, the sentence could be written as "Our data suggest that storms enhance mobilization of fresh leaf litter and shallow soils via increased surface runoff and shallow landslides.

**Response:** As suggested by the reviewer, we have reformulated the sentence (lines 22-23)**

**Comment 8:** L24: "Diverging mobilization pathways" is quite unclear here. The following sentence suggests that suspended sediment and POC mobilization are both related to water stage. And 1-day antecedent precipitation and discharge are both related to runoff, so it is unclear how these reflect different mobilization pathways. Perhaps the authors could interpret their statistical results to provide a mechanistic explanation of how 1-day antecedent moisture conditions affect POC in a different way than suspended sediment, as well as why discharge and suspended sediment concentration are more strongly linked than discharge and POC content. Given the amount of data included in this study, I would expect more robust interpretations of the data than what is highlighted here in the abstract. Are there any events captured in the dataset that can be analyzed to explain the effects of high runoff or landslides on POC export?

**Response:** This time series was not designed to investigate individual storm-driven flooding events or to tie POC export to specific geomorphological occurrences such as landslides. The field laboratory was intended to capture potential patterns across a range of flow conditions including storm events and baseflow conditions, and broadly infer potential mobilization and transport mechanisms of sediment and POC. Compositional, isotopic, and statistical evidence suggest that sediment and POC differ in their preferential sources and pathways in relation to discharge conditions. Particulate organic carbon exported during storm-driven events is characterized by more homogeneous stable carbon isotopic compositions and is enriched in 14C than in comparison to samples collected during baseflow conditions. This behavior suggests the tapping of an organic carbon source that is preferentially exported during storms such as coarse plant debris that requires a certain runoff strength to be mobilized. Daily precipitation (including 1-day and 2-day antecedent precipitation) is a measure of the intensity of rainfall and can be interpreted as a potential driver for particle transport to adjacent aquatic systems. Precipitation and discharge are related to runoff. However, both parameters are not interchangeable as they behave differently in response to the duration of rain events, spatial shifts in precipitation strength, catchment morphology, and soil water saturation/soil drainage. In the Sihl River watershed, we observe a

low correlation between daily discharge and daily precipitation (r = 0.30, p < 0.001, n = 17167; 1-day antecedent precipitation: r = 0.06, p < 0.001, n = 17167). Similarly, discharge and water stage are not linearly correlated as the rise of the water level will decelerate as soon as the water level reaches river banks and the adjacent riparian zone. Water stage might therefore act as a proxy for inundation and the mobilization of organic carbon deposited in the floodplain. Based on these assumptions, the relationships between water stage, 1-day antecedent precipitation, and POC concentrations might suggest that flooding, precipitation-induced erosion, and potentially shallow landsliding are major mechanisms facilitating the export of coarse discrete organic carbon via detaching litter and surface soil. The dependence of sediment concentrations on water stage and discharge might indicate that the majority of carbon-poor sediment is likely sourced from riverbeds and banks.

We acknowledge that the provided discussion is inadequate to fully convey our interpretation regarding potential mobilization drivers and pathways and has been revised in the abstract and expanded in section 4.3.

**Comment 9: L113: It is better to report discharge in common units of m3 /s**

**Response:** Discharge values for alpine streams are often reported in L s-1 (e.g., Smith et al., 2013; Turowski et al., 2016; Hilton et al., 2021). As these values are quoted in the manuscript, we utilized the unit given in the references. However, to further clarity for the reader, we have changed the units to "0.038 to 0.077 m3 s-1" in the manuscript (line 111).

**Comment 10:** L115: I suggest writing "...while bedrock lithologies in the ... are predominantly calcareous sandstones" rather than "bedrocks"

Response: Thank you for this suggestion. "Bedrocks" in line 113 are exchanged with "bedrock lithologies".

**Comment 11:** L122: Why was this sampling location chosen? Based on the coordinates provided, this stretch of river is heavily engineered, which may be both good and bad for this study. The channelization of the river may lead to efficient export of sediment from the upper catchment, however, this site is downstream of several dam-like structures that can retain sediment. How are the collected samples influenced by river engineering? Is there a portion of POC that could be missing due to sediment retention behind dams? Is river discharge also regulated by these dams and weirs, such that storm pulses are dampened and the effects of storms on POC export not fully captured? These issues should be addressed either here in the sample collection section or in the discussion section.

**Response:** The Sihl River time series was established as a field laboratory proximal to ETH Zurich. The sampling location in Allmend Park was chosen based on its accessibility, safety, and positioning within the watershed. Allmend Park can easily be reached from ETH within 20 minutes via public transport. The proximity of the sampling location further allowed a quick response time to episodic storm-driven events. Large segments of the Sihl river course are inaccessible. The location in Allmend Park is situated near (upstream of) the confluence with the Limmat River (~2 km) and thus captures the majority of the Sihl River watershed, chosen as intermediary between headwater streams and lowland rivers. Furthermore, the sampling site is located well downstream of significant headwater streams ensuring sufficient mixing of their suspended load.

The flow of the lower Sihl River is regulated by weir structures and run-off-the-river hydroelectric systems. These artificial barriers do not rise above channel banks leading to only minor increases in the hydraulic head. The resulting impoundments have little to no water storage capacity, with little potential to mitigate flooding. We believe that these man-made structures have a minimal impact on the export of water, sediment, and POC, while partly fortified banks and narrow floodplains likely aid the

downstream transport of the suspended load. Refer for a more detailed discussion to comment 2. We now address river engineering in sections 2.1 and 4.4.

**Comment 12:** L129-130: What is meant by identical replicates? Did the authors fill three separate sampling containers for each sample? Was each water sample filtered with three separate glass fiber filters? If there were three separate filters, how was it ensured that the sampled water was homogeneous with respect to sediment concentration? Please clarify what is meant by replicates.

**Response:** Thank you for pointing out that the collected samples were wrongly addressed as replicates. We passed the same sample of water over three separate glass fiber filters and averaged water and sediment concentrations to obtain a more robust representation of the suspended load. We have rephrased sentences in lines 125 and 129-130.

Comment 13: L133: were the DOC samples stored at 4 degrees C? Clarify the term "cooled."

**Response:** The reviewer's assumption is correct. DOC samples were stored at 4°C. We have revised the text to include the precise storage temperature (line 131).

**Comment 14:** L140-143: Were there no process blanks run to correct for possible carbon contamination introduced through filtration and sample handling? It is good practice to run process blanks in geochemistry.

Response: Several process blanks have been run and are incorporated in the evaluation of constant contamination.

**Comment 15:** L147: How was the CO2 captured and stored? Were CO2 sample vessels evacuated sufficiently before collecting evolved CO2? Was there a process blank for the sodium persulfate addition and helium purging steps? L145: Were DOC concentrations measured? If yes, add this procedure to the methods section. If not, how did the authors know how much carbon was in their aliquots?

**Response:** We used a wet chemical approach evolving and capturing CO2 directly in precombusted 12 mL gas-tight exetainer vials. Samples were measured within hours to 1-2 days, minimizing periods of storage. Sample vessels do not require evacuating as the ambient air and inorganic carbon are purged with helium prior to converting dissolved organic carbon to CO2.

Due to the relatively low dissolved organic carbon concentrations in the Sihl River, sample material was accumulated in an exetainer vial by repeated freeze-drying. The dissolved organic carbon was then reconstituted in Milli-Q water, oxidized using an acidified sodium persulfate solution (100 mL  $H_2O + 4.0$  g  $Na_2S_2O_8 + 200 \mu L$  of 85 %  $H_3PO_4$ ), and purged with high-purity helium gas (Grade 5.0, 99.9999% pure, for 10 min) removing ambient air and inorganic CO2. The sample was heated to 100°C for 1 h converting dissolved organic carbon to CO2 within the exetainer vial. Within 1 to 2 days, the exetainer vial was loaded into the carbonate handling system of the mini radiocarbon dating system (MICADAS, Ionplus) equipped with a gas-accepting ion source. The automated headspace sampling transfers the evolved CO2 over a magnesium perchlorate trap onto a zeolite molecular sieve. Subsequently, the zeolite trap is heated and the released CO2 is collected in a gas-tight syringe and diluted with helium. The He-CO2 mixture is then continuously pressed onto a Ti target in the gas-accepting ion source. Please, refer for a detailed description to Lang et al. (2012, 2016) and Wacker et al. (2013). In addition to phthalic acid and sucrose standards, we measured process, Milli-Q water, and persulfate blanks which are considered in the correction for extraneous carbon.

Dissolved organic carbon concentrations were measured and are reported in Appendix A and Table S1. Samples masses reported in line 145 are obtained from the elementar analyzer (Vario Micro, Elementar) coupled to the MICADAS system.

**Comment 16:** L156: Start this paragraph by saying what MixSIAR was used for and why before defining what the model is and how it was parameterized.

**Response:** We now provide an introductory sentence presenting our aim to apply MixSIAR as a Bayesian modeling approach to separate the contributions of potential organic carbon sources of the particulate load (lines 157-158)

**Comment 17:** L165: What outputs were used from the MixSIAR model and how? Did the authors use the full posterior distbrutions in the analysis, the median, the mean and standard deviations of the posterior distributions? Please clarify and explain how the model output was interpreted.

**Response:** As indicated in line 355 we reported the mean and standard deviation of the full posterior probability distributions and used the mean to calculate fluxes of endmember contribution of POC (lines 469-472).

**Comment 18:** L169: What is meant by "discern" here? This sentence is unclear. Do the authors mean that particulate matter concentrations tend to correlate with water discharge? If so, please be more clear.

**Response:** This sentence is more ambiguous than intended. River discharge is a key parameter determining the export of sediment and POC. We have adjusted the sentence accordingly (line 178).

Comment 19: L169: Where does the discharge data come from?

**Response:** Discharge data were retrieved from the gauging station Sihlhölzli, Zurich, operated by the Swiss Federal Office for the Environment (FOEN, https://www.hydrodaten.admin.ch) (lines 198-190). To avoid repetitions, we address predictor variables after listing all model approaches.

Comment 20: L170: by "insufficient" do the authors mean too infrequent? Be more specific.

Response: We agree with the reviewer and replace "insufficient" with "too infrequent" (line 179).

**Comment 21:** L175-177: Is there an equation for the bias correction factor? Is the correction factor a non-linear least squares regression approach? The last sentence is not well-integrated with the previous sentence. How does the correction affect the results? Is underestimation reduced or completely eliminated?

**Response:** The equation for the Duan bias correction factor ( $\varepsilon$ ) which removes the gross bias in estimates has been included in the manuscript:

$$\varepsilon = \frac{1}{N} \sum_{i=1}^{N} \exp\left(\ln\left(\mathcal{C}(obs)\right) - \ln(aQ^{b})\right)$$

with *C* as the observed suspended sediment concentration (mg L-1), *Q* as the water discharge (m3 s-1), and the rating coefficients *a* and *b*.

Commonly, discharge and sediment or POC concentrations are described by a power relationship. For the purpose of data fitting and the application of ordinary least squares regression, training data are transformed using natural logarithms to convert the model to its **linear** equivalent. Ordinary least squares function fits a linear function by minimizing the sum of the squares of the residuals. However, ordinary least squares in a log-log space introduces a systematic bias to the residuals. The retransformation into a power function thus often results in the overestimation of small values and the underestimation of large values. The Duan correction factor merely minimizes this bias. **Non-linear** least squares regression allows us to fit a power law function directly to the data without the need for logarithmic transformation and thus we don't encounter the bias of log-transformed residuals. But this approach also poses statistical problems as the assumption of homoscedasticity is often not met. We have carefully revised section 2.5.

**Comment 22:** L179: Introduce why machine learning techniques are applied here. It is unclear why fluvial loads need to be estimated when there is likely a gauging station with hydrologic data that can be paired with samples. Please provide an explanation of what the authors expect to achieve with the machine learning analyses and why different types of analyses need to be compared.

**Response:** Machine learning techniques are applied to estimate annual sediment and POC fluxes for the Sihl River basin (**line 186**). Power rating curves are limited in predicting export fluxes as they rely largely on the relationship between discharge and the suspended load. Machine learning approaches allow the application of more sophisticated functions - multilinear, support vector, random forest, and neural network regressions – and the utilization of additional predictor variables such as water stage, precipitation, 1-day, and 2-day antecedent precipitation to predict sediment and POC export. Utilized supervised machine learning approaches outperform traditional approaches in predicting sediment and POC concentrations (compare section 3.4-5; Figure 4, B1-2; Table B1-2). As none of the applied functions succeed in comprehensively describing the natural variations observed in the Sihl River basin (Table B1), we report the minimum and maximum values for the modeled annual loads as well as best-fitting model results throughout the manuscript.

We used the same training dataset for linear, non-linear, and machine learning functions consisting of observed sediment (n = 94) and POC (n = 90) concentrations and their respective discharge values obtained from the Sihlhölzli gauging station operated by the Swiss Federal Office for the Environment (FOEN, https://www.hydrodaten.admin.ch). The machine learning training dataset is augmented with water stage (FOEN), precipitation, 1-day, and 2-day antecedent precipitation values (Federal Office of Meteorology and Climatology, MeteoSwiss, https://gate.meteoswiss.ch).

**Comment 23:** L187-189: Why were these predictor variables chosen? Are there other studies that have shown these to be statistically useful parameters for predicting river particulate matter fluxes? If yes, please cite these references here.

**Response:** Studies showed that utilized parameters are statistically useful for predicting river particulate matter fluxes and have been listed in lines 181 to 184. We have included these references in line 203.

Our results also indicate that usage of water stage, precipitation, and 1-day antecedent precipitation are critical variables enhancing the performance of model approaches in predicting sediment and organic carbon concentrations (Figure 4, B1, and B2; Table B1). Soil moisture as a parameter for soil water saturation might also pose as a useful variable. However, satellitebased soil moisture data are short (2015-now), sparse (2-3-day returns), and patchy (missing data) to form a continuous record needed to estimate robust annual yields for sediment and POC loads. **Comment 24:** L320-324: Based on the data plotted in Figure 5, it appears that there is either an older soil OC source or aquatic biomass source. The authors need to propose a mechanism for producing the isotope values that do not plot between their vegetation, soil, and bedrock endmembers. If they are going to rule out autochthonous production, then they need to provide an alternate hypothesis.

**Response:** Endmember compositions plotted in Figure 5a do not represent the full range of observed values but consist of the mean and one standard deviation (Table 2). Observed stable isotopic compositions for soils span values from -31.36 to -22.70 ‰ (Gies et al., 2022; Smith et al., 2013; van der Voort et al., 2016). Similarly, the vegetation endmember ranges from -31.37 to - 22.70 ‰ (Gies et al., 2022; Smith et al., 2013). Based on these values, Sihl River samples fall within the proposed endmember compositions. The isotopic compositions of Lake Sihl samples are discussed in section 4.2. We have replaced the combined vegetation endmember with the original foliage and wood endmembers (Figure B).

Endmember compositions are limited by small sample sizes and are associated with large uncertainties how well these samples represent the vegetation and soils present in the watershed. The isotopic composition of the Erlenbach, Lümpenbach, and Vogelbach indicate vegetation, soil, and bedrock sources that are not captured in previous surveys. Another potential source might consist of carbon-poor bank and riverbed sediments. We assume aquatic primary productivity in headwater streams to be negligible due to limited light conditions/shading, steep gradients, turbulent flow, and short residence times.